# Insulin post-transcriptionally modulates Bmal1 protein to affect the hepatic circadian clock

Fabin Dang[1,2,*], Xiujie Sun[1,2,*], Xiang Ma[1,2], Rong Wu[1,2], Deyi Zhang[1,2], Yaqiong Chen[1], Qian Xu[3], Yuting Wu[1] & Yi Liu[1,2]

Although food availability is a potent synchronizer of the peripheral circadian clock in mammals, the underlying mechanisms are unclear. Here, we show that hepatic Bmal1, a core transcription activator of the molecular clock, is post-transcriptionally regulated by signals from insulin, an important hormone that is temporally controlled by feeding. Insulin promotes postprandial Akt-mediated Ser42-phosphorylation of Bmal1 to induce its dissociation from DNA, interaction with 14-3-3 protein and subsequently nuclear exclusion, which results in the suppression of Bmal1 transcriptional activity. Inverted feeding cycles not only shift the phase of daily insulin oscillation, but also elevate the amplitude due to food overconsumption. This enhanced and reversed insulin signalling initiates the reset of clock gene rhythms by altering Bmal1 nuclear accumulation in mouse liver. These results reveal the molecular mechanism of insulin signalling in regulating peripheral circadian rhythms.

[1] Key Laboratory of Nutrition and Metabolism, Institute for Nutritional Sciences, Shanghai Institutes for Biological Sciences, Chinese Academy of Sciences, 320 Yueyang Road, Shanghai 200031, China. [2] Shanghai Institutes for Biological Sciences, University of Chinese Academy of Sciences, Shanghai 200031, China. [3] Department of Endocrinology, The First Affiliated Hospital of Harbin Medical University, No. 23 Youzheng Street, NanGang District, Harbin 150001, China. * These authors contributed equally to this work. Correspondence and requests for materials should be addressed to Y.L. (email: liuyi@sibs.ac.cn).

In mammals, the approximate 24 h oscillation of the circadian clock is critical for maintaining metabolic homoeostasis. A pacemaker in the suprachiasmatic nucleus (SCN) of the hypothalamus functions as a central clock to lead an orchestra of individual rhythms in peripheral tissues[1]. However, this dominance of the SCN in synchronizing subordinate organs is overridden by food availability, as modification of the feeding cycle by food entrainment resets subsidiary clocks but has little effect on the master clock[2–4]. Although numerous efforts have been undertaken to reveal the molecular basis of feeding-sensitive synchronizer(s), precise molecular mechanisms are unclear.

Cyclic signals from both clocks are generated by a well-defined autonomous oscillator that consists of two coupled negative-feedback loops. At the centre of this molecular machinery are heterodimers of the two transcription activators BMAL1 and CLOCK that stimulate the expression of repressors for their own activity (Per and Cry) and transcription (Nr1d1) (refs 5–8). Surrounding this core clockwork, many signalling molecules can influence its output, including nuclear receptors (glucocorticoid receptor)[9], transcription activators/coactivators (PGC-1α)[10], redox sensors (SIRT1) (refs 11,12) and energy sensors (AMPK)[13].

Tightly associated with feeding status, the temporal signalling of fasting and refeeding is considerably altered during food entrainment. Thus it comes as no surprise to speculate that these signals are involved in the regulation of the peripheral clock. Several lines of evidence support this notion. As the phases of core clock genes in peripheral tissues are reset by the activation of glucocorticoid receptor[9], fasting supposedly affects the peripheral clock by the elevation of the plasma glucocorticoid levels[14]. In addition, glucagon-CREB/CRTC2, one of fasting signalling cascades, modulates Bmal1 expression in mouse liver[15]. Moreover, PGC-1α regulates the hepatic expression of Bmal1 (ref. 10), on the other hand, fasting promotes the transcription of Pgc-1α via the glucagon-CREB signalling pathway in the liver[16], which is repressed by refeeding[17]. In contrast to these well-defined roles of fasting hormones in the regulation of the peripheral clock, how the hepatic clock senses the key refeeding hormone insulin has not been thoroughly identified, although it has been implicated in modulating the peripheral clock[18–21]. Here, we show that insulin reduces Bmal1 transcriptional activity by affecting its intracellular localization in mouse liver, which requires Akt-dependent phosphorylation of the Ser42 residue. Our results demonstrate that this signalling mechanism has a central role in the initial phase during food entrainment resetting of the hepatic clock.

## Results

**Insulin modulates hepatic Bmal1 nuclear accumulation.** Previous reports from several groups have observed 8–12 h phase advances of Bmal1 mRNA rhythm before its target gene expression (Dbp and Nr1d1) in mouse liver[22–24]. Such unusually long delay between the expression of a transcription factor and its regulated genes suggests that Bmal1 transcriptional activity may be post-transcriptionally modulated. This notion is further supported by the fact that the fluctuations of Bmal1 nuclear accumulation and DNA-binding activity are uncoupled from its total protein oscillation in the liver[22,24,25], whereas the underlying mechanism remains elusive. We observed consistent results that total Bmal1 protein amounts peaked around 8 h ahead of its nuclear levels (delayed from ZT20 to ZT4) and 12 h before its binding with Dbp promoter and subsequent induction of Dbp expression (delayed from ZT20 to ZT8) in livers of ad libitum fed mice (Supplementary Fig. 1a–c). Interestingly, we noticed that Bmal1 protein contains three consensus recognition motifs (at Ser 42, 422 and 513) for Ser/Thr kinase Akt, a key mediator of insulin

signalling (Supplementary Fig. 1d). Considering that insulin has been reported to affect clock gene expression[26,27] and its circulating levels also peaked around ZT20 (Supplementary Fig. 1e) when the oscillation of nuclear Bmal1 proteins reached the trough, we wondered whether insulin may block Bmal1 nuclear accumulation even with the peak of its total protein levels (Supplementary Fig. 1f). Indeed, Bmal1 protein amounts were significantly reduced in the nuclei but increased in the cytosol by insulin treatment in human hepatoma HepG2 cells or primary hepatocytes, as demonstrated by immunoblotting or immunostaining results (Fig. 1a,b), whereas insulin had little effect on the nucleus-cytosol shuttling of other key clock members (Clock, Cry1 and Per1, Fig. 1a). Moreover, the immunostaining results also showed that leptomycin B (LMB), a specific nuclear protein export inhibitor, abolished the inhibitory effect of insulin on nuclear Bmal1 accumulation (Fig. 1b), which was further verified by cell fraction analysis (Fig. 1c). Altogether, these data demonstrate that insulin modulates the subcellular localization of Bmal1 proteins in vitro.

We then examined whether insulin exerts the similar effect on hepatic Bmal1 in vivo. As mice usually do not eat much and their plasma insulin levels arrive at the nadir during light time, we fasted mice from ZT0 to prevent them from eating to disturb the results, and then intraperitoneally injected these animals with insulin at ZT6 and killed them after 0.5 to 6 h as indicated in Supplementary Fig. 1g. As expected, insulin markedly reduced nuclear Bmal1 protein accumulation, correspondingly increased its cytosolic distribution, and mildly enhanced its total amounts in mouse liver (Fig. 1d). As a result, insulin injection significantly attenuated Bmal1 occupancy on the promoters of its target genes (Dbp and Nr1d1, Fig. 1e) and subsequently their transcription (Fig. 1f). In line with previously reported in vitro results[19], we also found that insulin injection increased hepatic mRNA levels of Per1 and Per2, the other two Bmal1-target genes (Supplementary Fig. 1h). As Bmal1 deficiency has been shown to enhance the expression of these two genes in mouse liver[23,24], our data strongly indicate that insulin suppresses Bmal1 transcriptional activity. Meanwhile, Bmal1 expression was slightly reduced by insulin injection in the early stage of this experiment, whereas significantly increased at ZT12 (Supplementary Fig. 1g), which may be caused by the decrease of Nr1d1 that feedback inhibits Bmal1 transcription[8]. By contrast, no obvious difference in nuclear Bmal1 amounts and Dbp expression was observed in livers collected from the similar experiments performed during dark time (Supplementary Fig. 1i,j), when circulating insulin levels are high. Moreover, the inhibitory effect of insulin on Dbp expression was abolished in Bmal1 null primary hepatocytes (Fig. 1g), suggesting such insulin effect is mediated by Bmal1. In summary, our results indicate that insulin affects Bmal1 intracellular localization to limit its transcriptional activity in mouse liver.

**Akt phosphorylates Bmal1 at Ser42 residue.** As mentioned above that Bmal1 contains three consensus recognition motifs for Akt (Supplementary Fig. 1d), we wondered whether Akt is capable of phosphorylating Bmal1, which may be involved in insulin regulation of Bmal1. As expected, exposure of primary hepatocytes to insulin increased the phosphorylation of Bmal1, as detected by immunoblot assay with a phospho-Akt substrate antiserum (Fig. 2a), which was diminished in primary hepatocytes with the deficiency of Akt2, the major Akt isoform expressed in the liver (Fig. 2b). In addition, the results of co-immunoprecipitation assay showed the direct interaction between Bmal1 and Akt2 (Fig. 2c). We then performed liquid chromatography tandem mass spectrometry to

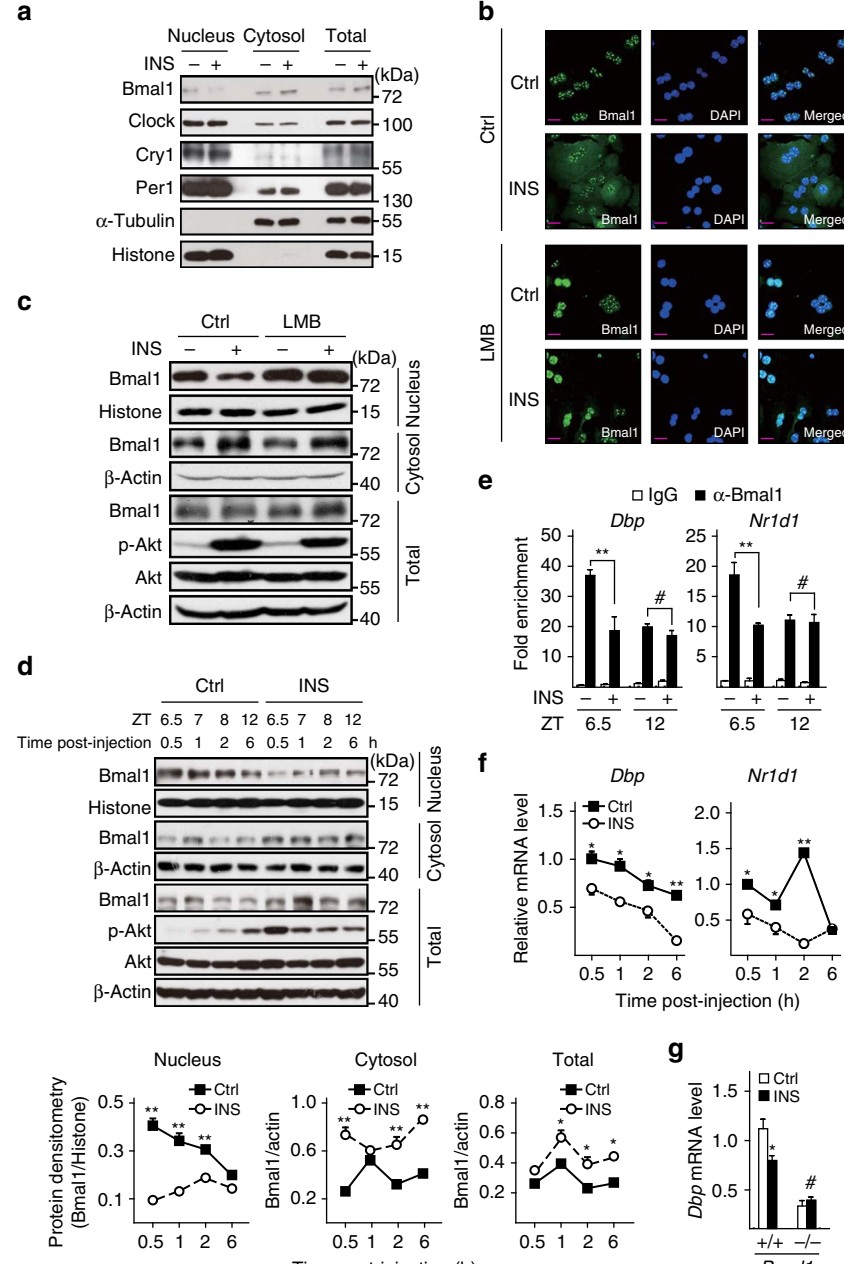

**Figure 1 | Insulin regulates hepatic Bmal1 translocation and activity.** (**a**) Immunoblotting analysis of endogenous Bmal1, Clock, Per1 and Cry1 proteins in nuclear, cytosolic and total extracts of HepG2 cells treated with insulin (INS, 50 nM) or control vehicle for 30 min. (**b**) Immunostaining analysis of endogenous Bmal1 localization in primary hepatocytes in the presence or absence of LMB (20 ng ml$^{-1}$, 5 h pretreatment) or insulin (INS, 50 nM, 30 min). Bmal1 staining (left), DAPI staining (middle) to visualize nuclei, and merged together (right). A short purplish red line indicates a scale bar = 50 µm. (**c**) Immunoblotting analysis of endogenous Bmal1 protein in nuclear, cytosolic and total extracts of primary hepatocytes treated with LMB (20 ng ml$^{-1}$, 5 h pretreatment) or insulin (INS, 50 nM, 30 min). Mice were fasted from ZT0 and injected intraperitoneally with insulin (INS, 2 U kg$^{-1}$) or normal saline at ZT6, and then animals were killed at indicated intervals (**d**–**f**, $n = 3$). (**d**) Immunoblotting analysis of endogenous Bmal1 protein in pooled nuclear, cytosolic and total extracts of livers (top), and corresponding densitometry analysis of relative Bmal1 protein amounts were shown at the bottom; (**e**) ChIP analysis of the occupancy of Bmal1 on *Dbp* and *Nr1d1* promoters in livers from mice collected at ZT6.5 or ZT12 (0.5 or 6 h post-injection, respectively); (**f**) quantitative PCR analysis of hepatic mRNA levels of *Dbp* and *Nr1d1*. (**g**) Quantitative PCR analysis of mRNA levels of *Dbp* in insulin-treated (INS, 50 nM, 1 h) primary hepatocytes from WT (+/+) or *Bmal1* liver-specific knock-out (−/−) mice ($n = 3$). Data are represented as mean ± s.e.m, statistical analyses were performed with a two-tailed unpaired Student's *t*-test, *$P < 0.05$, **$P < 0.01$, #no significant difference.

characterize residue(s) in Bmal1 that undergo insulin-induced phosphorylation. This analysis revealed one single site Ser42 on Bmal1 (Supplementary Fig. 2a), which is highly conserved in vertebrates (Supplementary Fig. 2b). Consistent with mass spectrometry results, constitutive active Akt2 (Akt2-CA) induced wild-type (WT) Bmal1 protein phosphorylation, which was abolished by a serine to alanine mutation at Ser42 residue (S42A) but not a S422A/S513A double mutation, as shown by immunoblot assay with phospho-Akt substrate antiserum (Fig. 2d). Similarly, exposure of hepatocytes to insulin increased the phosphorylation of WT Bmal1, but not the S42A mutant (Supplementary Fig. 2c,d).

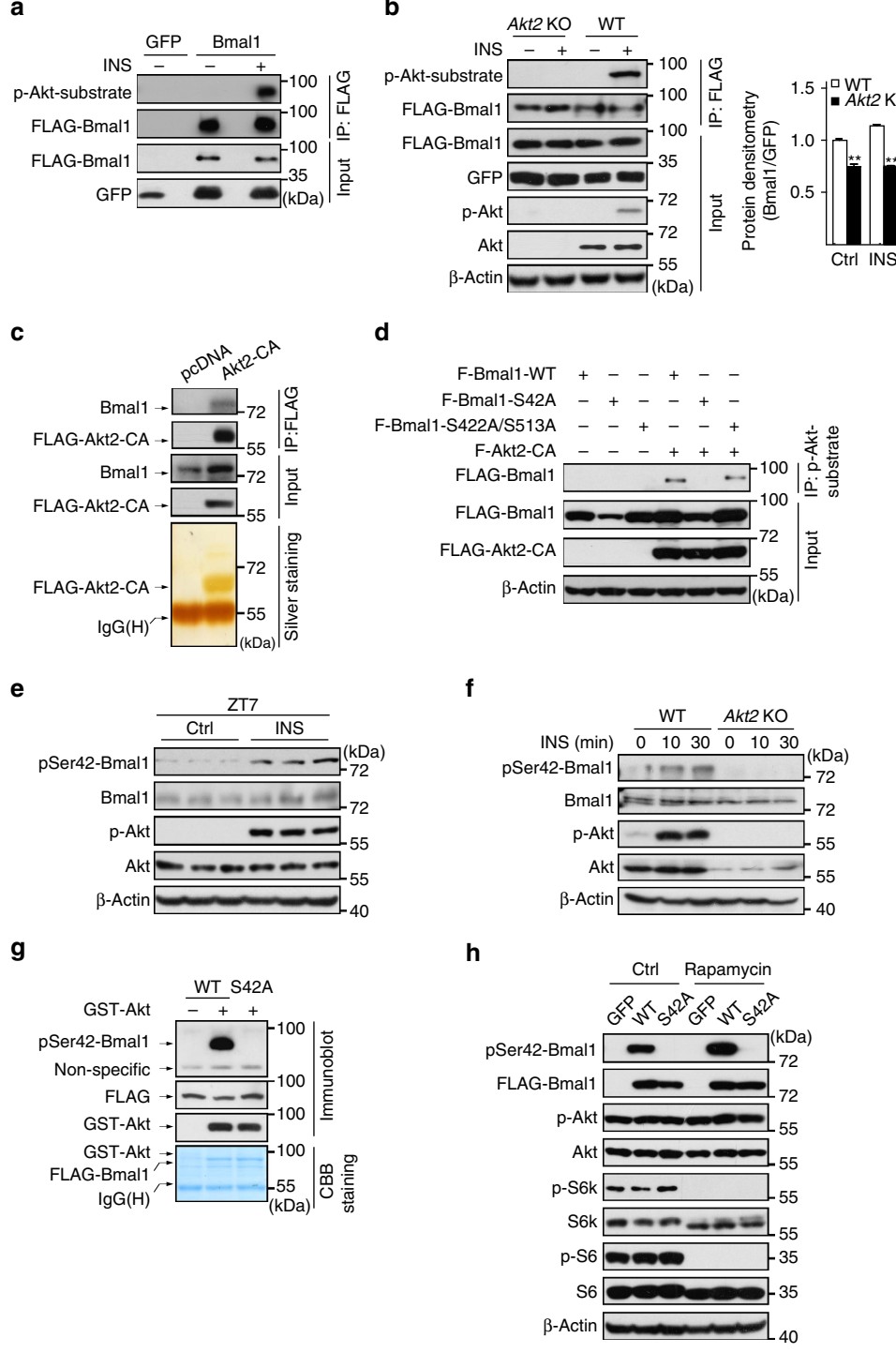

**Figure 2 | Insulin promotes Akt-mediated phosphorylation of Bmal1 at Ser42.** Immunoblotting analysis of proteins recognized by phospho-Ser/Thr-Akt substrate antiserum in FLAG immunoprecipitates (IP) prepared from (**a**) primary hepatocytes infected with Ad-FLAG-Bmal1 or Ad-GFP, (**b**) primary WT or *Akt2* null (*Akt2* KO) hepatocytes infected with Ad-FLAG-Bmal1 in the presence or absence of insulin (INS, 50 nM, 30 min). (**c**) Immunoblotting analysis of endogenous Bmal1 proteins co-immunoprecipitated by anti-FLAG antibody from lysates of HEK 293T cells transfected with FLAG-Akt2-CA (top), the corresponding gel was silver-stained (bottom). (**d**) Immunoblotting analysis of FLAG-Bmal1 proteins recognized by FLAG-HRP antiserum in phospho-Ser/Thr-Akt substrate immunoprecipitates prepared from HEK 293T cells transfected with indicated plasmids. (**e**) Immunoblotting analysis of pSer42-Bmal1 protein amounts in liver homogenates from mice fasted from ZT0, injected intraperitoneally with insulin (INS, 2 U kg$^{-1}$) at ZT6 and then killed at ZT7. (**f**) Immunoblotting analysis of pSer42-Bmal1 protein amounts in primary WT or *Akt2* null (*Akt2* KO) hepatocytes exposed to insulin (INS, 50 nM) for indicated times. (**g**) *In vitro* kinase assay of Bmal1 protein phosphorylated by active recombinant Akt in immunoprecipitates of WT or Ser42Ala mutant (S42A) FLAG-Bmal1 from HEK 293T cell lysates. Ser42-phosphorylated Bmal1 (pSer42-Bmal1) was detected by phospho (Ser42) specific Bmal1 antiserum, the corresponding gel was Coomassie Brilliant Blue-stained (CBB, bottom). (**h**) Immunoblotting analysis of Ser42-phosphorylated Bmal1 (pSer42-Bmal1) protein amounts in Ad-GFP, Ad-FLAG-tagged WT or S42A mutant Bmal1 virus-infected primary hepatocytes treated with insulin (50 nM, 30 min) in the presence or absence of rapamycin (100 nM, 1 h pretreatment).

To specifically detect the phosphorylation of Ser42 on Bmal1, we generated a Ser42-phosphorylation-site-specific antibody (pSer42-Bmal1) with the epitope of $R_{37}KRKGSATDYQE_{48}$. We first confirmed its specificity by competitive inhibition studies with phospho or non-phosphopeptides in immunoblotting assays (Supplementary Fig. 2e). By using this antibody, we found that insulin significantly stimulated Ser42-phosphorylated Bmal1 accumulation in the liver (Fig. 2e and Supplementary Fig. 2f) as well as WT primary hepatocytes, which was markedly reduced in primary hepatocytes isolated from *Akt2* knock-out mice or treated with PI3K inhibitor, LY-294002 (Fig. 2f, Supplementary Fig. 2g). The effect of Akt on Bmal1 appears direct as recombinant Akt protein derived from Sf9 cells was able to phosphorylate WT but not S42A mutant Bmal1 *in vitro* (Fig. 2g).

As a previous work has reported that S6K kinase is also capable of phosphorylating Bmal1 on Ser42 (ref. 28), we then checked whether Akt is the key mediator for insulin induction of Bmal1-Ser42 phosphorylation. In contrast to expectedly suppressing insulin-promoted Ser42 phosphorylation of Bmal1 proteins, the inhibition of S6k kinase activity by rapamycin treatment actually enhanced the amounts of pSer42-Bmal1 (Fig. 2h), which may be due to the loss of the negative-feedback regulation of S6k on insulin receptor substrate proteins[29]. In summary, our data indicate that insulin stimulates Akt-mediated phosphorylation of Ser42 on Bmal1.

**S42 phosphorylation affects Bmal1 localization and stability**. As the above data suggest that insulin may limit nuclear Bmal1 accumulation via Akt-mediated Ser42 phosphorylation of Bmal1, we first verified whether Akt is involved in the regulation of Bmal1 subcellular distribution *in vivo* by checking the circadian pattern of nuclear Bmal1 protein in livers of *Akt2* null mice. As expected, *Akt2* deficiency significantly enhanced nuclear Bmal1 accumulations around ZT16–20, which results in the disruption of its nuclear rhythm (Fig. 3a and Supplementary Fig. 3a) and the increase of its target gene expression (*Dbp* and *Nr1d1*) at multiple time points (Supplementary Fig. 3b), even though these animals and WT littermates exhibited comparable rhythms of food intake (Supplementary Fig. 3c). Complementarily, no significant differences in nuclear Bmal1-S42A protein amounts were observed between the livers of *Akt2* null and WT mice killed around ZT20 (Supplementary Fig. 3d). In line with the *in vivo* results, overexpression of Akt2-CA strikingly lowered Bmal1 protein abundance in the nucleus in primary hepatocytes (Fig. 3b and Supplementary Fig. 3e).

It is well established that AKT recognition sequences often overlap with putative binding sites of 14-3-3 proteins that affect subcellular localization of AKT-substrate proteins via a direct association[30,31]. Thus, we wondered whether phosphorylated Bmal1 interacts with 14-3-3. Indeed, insulin significantly enhanced the amounts of 14-3-3β proteins recovered from primary hepatocyte lysates by an anti-Bmal1 antibody, which was diminished by the pretreatment of these lysates with lambda phosphatase (Fig. 3c). Consistently, intraperitoneal injection of insulin enhanced the interaction between Bmal1 and 14-3-3β in the liver (Fig. 3d). Moreover, Bmal1-S42A mutation diminished insulin-induced Bmal1-14-3-3β interaction in primary hepatocytes (Fig. 3e), suggesting the necessity of Ser42 phosphorylation in this regards. Interestingly, the Akt/14-3-3-recognition motif of Ser42 ($R_{37}KRKG\mathbf{S}_{42}$) overlaps with one of previously characterized nuclear localization signal motif ($N_{36}RKRKG_{41}$) on Bmal1 (ref. 32), which implies that Ser42 phosphorylation and subsequent interaction with 14-3-3β may affect Bmal1 localization. Supporting this notion, Bmal1-S42A mutant proteins resisted to the nuclear exclusion promoted by

insulin treatment (Fig. 3f) or Akt2-CA overexpression (Fig. 3g and Supplementary Fig. 3f), compared with its WT proteins in primary hepatocytes. Taken together, these data suggest that insulin-Akt-14-3-3 signalling pathway plays a pivotal role in modulating Bmal1 subcellular distribution, which requires the phosphorylation of its Ser42 residue.

During studying the effect of insulin signalling on Bmal1 subcellular localization, we also noticed that total Bmal1 protein amounts were enhanced by insulin injection (Fig. 1d), but reduced by *Akt2* deficiency (Fig. 3a and Supplementary Fig. 2h), suggesting the involvement of insulin-Akt signalling in the regulation of Bmal1 protein stability. This notion was further supported by the *in vitro* results that S42A mutation unstabilized Bmal1 (Supplementary Fig. 3g), whereas Akt2-CA increased Bmal1-WT protein amounts in HEK 293T cells (Supplementary Fig. 3h). Since *Bmal1* deficiency has been reported to disrupt insulin-induced phosphorylation of Akt[33], we further confirmed the necessity of Bmal1 for insulin inhibitory effect on *Dbp* expression (Fig. 1g) by using an adenovirus of Akt2-CA. In the absence or presence of insulin treatment, overexpression of Akt2-CA expectedly reduced *Dbp* expression to the comparable levels in insulin-treated WT hepatocytes, but exerted little effect in *Bmal1* null hepatocytes (Supplementary Fig. 3i).

**Role of S42 phosphorylation in insulin modulation of Bmal1**. Next, we explored the effect of Akt-mediated Ser42 phosphorylation on Bmal1 transcriptional activity. With their PAS domains, Bmal1 and Clock form heterodimers that bind to E-box enhancer elements in the promoters of target genes by their basic helix-loop-helix domains[34]. Since Ser42 is close to the basic helix-loop-helix domain and far away from the two PAS domains on Bmal1, we hypothesized that phosphorylation of Ser42 may affect Bmal1 affinity to DNA but not Clock. Indeed, the results of chromatin immunoprecipitation assays showed that the amounts of Bmal1-S42A proteins recruited to E-box in the promoters of *Dbp* and *Nr1d1* were three to fourfold higher than those of WT Bmal1 (Fig. 4a, left), even with less nuclear protein amounts (Fig. 4a, right), which suggests that phosphorylation of Ser42 disrupts Bmal1 binding to DNA. By contrast, S42A mutation did not affect Bmal1 associated with Clock in the absence of insulin, as determined by co-immunoprecipitation (Co-IP) assay (Supplementary Fig. 3j). As a result, Bmal1 mutants containing S42A induced much higher *Per1*-Luc activity than that of WT Bmal1 with co-expression of Clock in HEK 293T cells and neither S422A nor S513A mutation exhibited similar effects (Fig. 4b), suggesting a potential role of Ser42 phosphorylation in suppressing basal transcriptional activity of Bmal1. Supporting this notion, adenoviral expression of Bmal1-S42A mutant significantly increased the mRNA levels of Bmal1-target genes (*Dbp* and *Nr1d1*) in primary hepatocytes treated with insulin, compared with Ad-Bmal1-WT (Fig. 4c). In line with these *in vitro* results, the insulin-induced reduction of hepatic protein amounts of Dbp and Nr1d1 was reversed in mice tail-vein injected with Ad-Bmal1-S42A viruses (Fig. 4d). Taken together, these results suggest the importance of Ser42 phosphorylation in modulation of Bmal1 transcriptional activity.

**Insulin initiates hepatic clock reset by food entrainment**. Food availability is a potent synchronizer for the peripheral clock, as modification of feeding cycles by food entrainment resets the hepatic clock but exerts little impact on master clock in the SCN[2–4]. Tightly associated with feeding status, the temporal signalling of insulin is considerably altered during food entrainment. It thus comes as no surprise to speculate that insulin signal is involved in the regulation of the peripheral clock

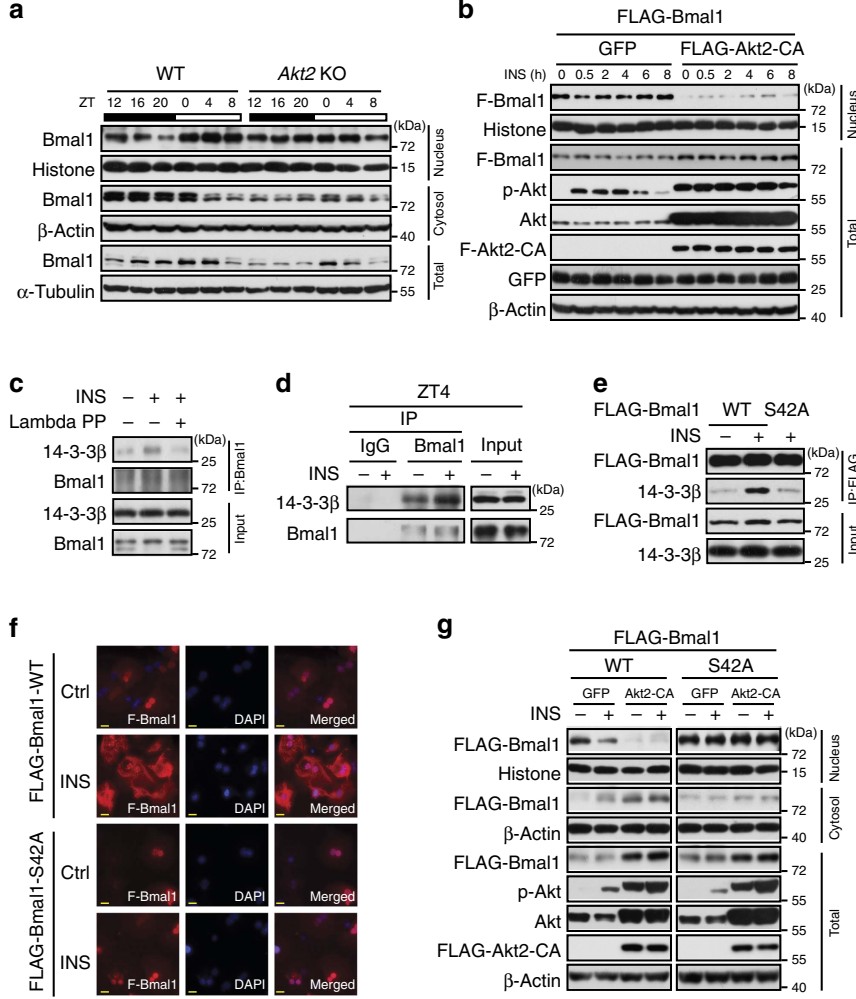

**Figure 3 | Insulin affects Bmal1 subcellular distribution and 14-3-3β association. (a)** Immunoblotting analysis of Bmal1 protein in pooled nuclear, cytosolic and total extracts of livers from *ad libitum* fed WT or *Akt2* KO mice killed at 4 h intervals around the clock (*n* = 3). **(b)** Immunoblotting analysis of FLAG-Bmal1 (F-Bmal1) in nuclear and total extracts of primary hepatocytes infected with adenoviruses expressing either GFP control or FLAG-Akt2-CA in the presence or absence of insulin treatment (INS, 50 nM) for indicated times. **(c)** Immunoblotting analysis of 14-3-3β protein amounts co-immunoprecipitated by the anti-Bmal1 antibody from control or lambda phosphatase treated lysates of primary hepatocytes in the presence or absence of insulin (50 nM, 30 min). **(d)** Immunoblotting analysis of 14-3-3β protein amounts co-immunoprecipitated by the anti-Bmal1 antibody from pooled lysates of mice fasted from ZT0, injected intraperitoneally with insulin (2 U kg$^{-1}$) at ZT2 and then killed at ZT4 (*n* = 3). **(e)** Immunoblotting analysis of 14-3-3β protein amounts co-immunoprecipitated by the anti-FLAG beads from lysates of insulin (50 nM, 30 min) treated primary hepatocytes infected with Ad-FLAG-tagged WT or S42A mutant Bmal1 adenoviruses. **(f)** Immunostaining analysis of FLAG-Bmal1-WT or S42A protein localization in primary hepatocytes infected with corresponding adenoviruses in the presence or absence of insulin (INS, 50 nM, 30 min). FLAG-Bmal1 staining (F-Bmal1, left), DAPI staining (middle) to visualize nuclei, and merged together (right). Scale bar = 50 μm. **(g)** Immunoblotting analysis of FLAG-Bmal1-WT or S42A protein amounts in nuclear, cytosolic and total extracts of primary hepatocytes infected with adenoviruses of GFP, FLAG-Akt2-CA, FLAG-Bmal1-WT or S42A in the presence or absence of insulin treatment (INS, 50 nM, 30 min).

from the beginning of the change of feeding times. This notion is supported by several reports[18–21], however, much less is known about the underlying mechanism. As the cycle of feeding periods is reversed from the start of daytime restricted feeding and we have shown the role of insulin signal in regulating Bmal1 transcriptional activity, we wanted to examine whether the dynamics of nuclear Bmal1 is altered correspondingly in mouse liver. Mice were killed every 4 h over the first 72 h under restricted feeding (RF) initiated at ZT12 or *ad libitum* feeding (AF) as control (Supplementary Fig. 4a). RF expectedly reversed the phase of nuclear Bmal1 protein oscillation from the beginning of Day1 and in the following 2 days (Fig. 5a and Supplementary Fig. 4b), which was disrupted in *Akt2* knock-out mice (Fig. 5b and Supplementary Fig. 4c). Consequently, RF initiated the phase shifting of Bmal1-target gene expression (*Dbp* and *Nr1d1*) from

Day1-ZT12 (Fig. 5c), whereas the phase of *Bmal1* mRNA rhythm had not been altered until Day2 (Fig. 5c), which is mainly due to the change of *Nr1d1* mRNA oscillation that inhibits *Bmal1* expression[8].

Interestingly, the peak of diurnal insulin oscillation was elevated gradually from an AF comparable level during Day1 RF (1.66 ± 0.13 ng ml$^{-1}$ at RF-Day1-ZT8 versus 1.66 ± 0.53 ng ml$^{-1}$ at AF-Day1-ZT20) to a strikingly high level in Day3 RF (10.01 ± 0.91 ng ml$^{-1}$ at Day3-ZT4, Fig. 5d). To understand how RF causes such high insulin levels, we measured food consumption in these animals. RF did not significantly change the daily food intakes during the first two days until Day3 (Supplementary Fig. 4d), suggesting that RF mice were trained to learn the food shortage during dark time and prepare for it by consuming much more amounts of food during the feeding

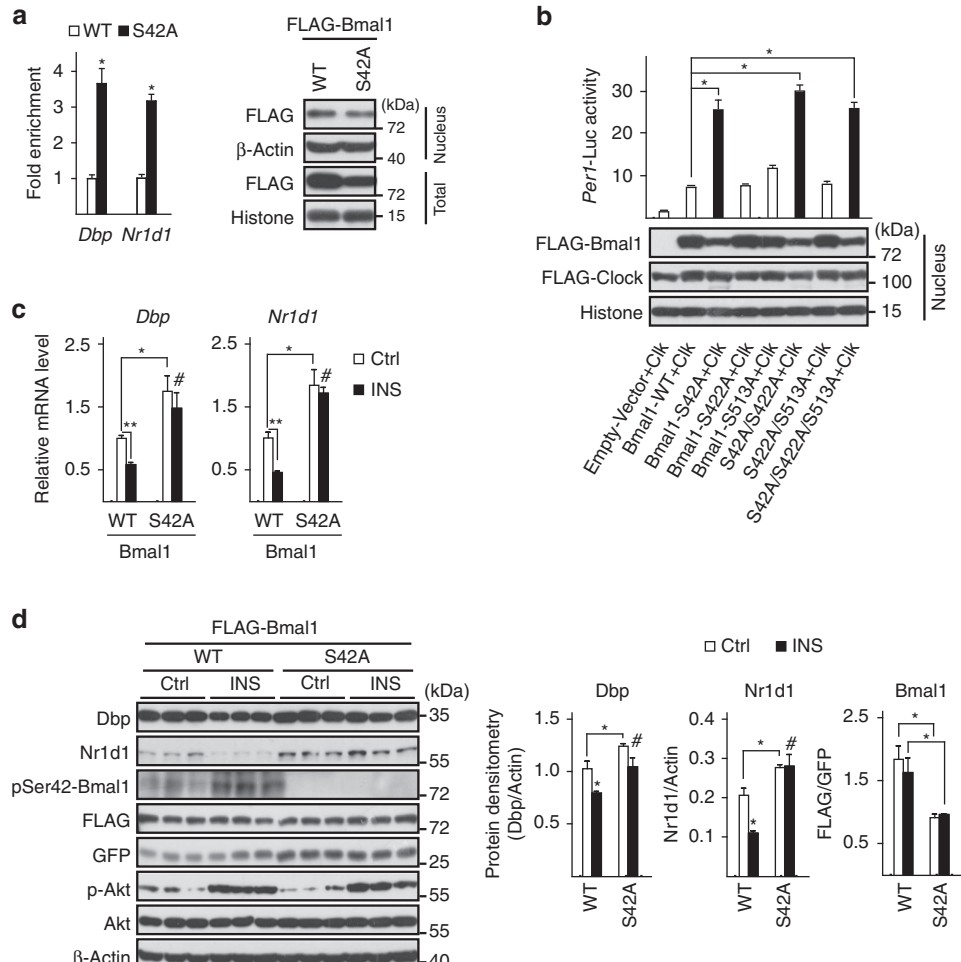

**Figure 4 | Ser42 phosphorylation affects Bmal1 transcriptional activity.** (**a**) ChIP analysis of the occupancy of Ad-FLAG-WT-Bmal1 or Ad-FLAG-S42A-Bmal1 on the promoters of *Dbp* and *Nr1d1* in primary hepatocytes co-infected with Ad-Clock (*n* = 3, left). Immunoblotting analysis of relative FLAG-Bmal1 (FLAG) protein amounts in experiments with the corresponding conditions (right). (**b**) Transient luciferase reporter assay of *Per1*-Luc activity in HEK 293T cells co-transfected with Clock together with control vector, WT or indicated Bmal1 mutants (top, *n* = 3). The amounts of nuclear FLAG-Bmal1 proteins in experiments with the corresponding conditions were shown by immunoblotting analysis at the bottom. (**c**) Quantitative PCR analysis of mRNA levels of *Dbp* and *Nr1d1* in insulin-treated (50 nM, 1h) primary hepatocytes infected with Ad-FLAG-WT-Bmal1 or Ad-FLAG-S42A-Bmal1 adenoviruses (*n* = 3). (**d**) Immunoblotting analysis of Dbp and Nr1d1 protein amounts in liver homogenates from mice tail-vein injected with Ad-FLAG-WT-Bmal1 or Ad-FLAG-S42A-Bmal1 and injected intraperitoneally with insulin (2 U kg$^{-1}$) at ZT6 and then killed at ZT8, and corresponding densitometry analysis of relative Dbp, Nr1d1 and FLAG-Bmal1 protein amounts were shown on the right (*n* = 3). Data are represented as mean ± s.e.m, statistical analyses were performed with a two-tailed unpaired Student's *t*-test, *$P < 0.05$, **$P < 0.01$, #no significant difference.

period than AF mice did. Supporting this notion, RF significantly increased the mouse stomach sizes and weights at ZT4, 8 and 12 in each day (Fig. 5e and Supplementary Fig. 4e) and reduced the amplitude of plasma leptin oscillation (Fig. 5f), compared with AF. By contrast, no significant change of the plasma levels of glucagon, which has been reported to affect *Bmal1* transcription[15], was observed at Day3-ZT16 and ZT4 (Supplementary Fig. 4f) when insulin levels were strikingly different between AF and RF mice, which may be due to a relatively short fasting period and food overconsumption during feeding times. Altogether, these data demonstrate a role of insulin in the regulation of the hepatic clock via modulating Bmal1 subcellular localization in food-entrained mice.

**Effect of Bmal1 S42 phosphorylation on the hepatic clock.** Next, we examined the importance of Ser42 phosphorylation in Bmal1 regulating the hepatic clock under either AF or RF conditions. As expected, the mRNA levels of *Dbp* and *Nr1d1* were significantly enhanced in livers collected around either ZT4 or

ZT20 from AF mice tail-vein injected with Ad-Bmal1-S42A, compared with those with Ad-Bmal1-WT (Fig. 6a). Consistently, hepatic Bmal1-S42A proteins resisted to feeding-induced nuclear exclusion around ZT4 and ZT8 in Day1-RF mice, compared with Bmal1-WT proteins (Fig. 6b and Supplementary Fig. 4g), which resulted in the enhancement of *Dbp* and *Nr1d1* expression during feeding times (ZT4–ZT8, Fig. 6c). In line with the *Akt2* knock-out data (Supplementary Fig. 3b), Ad-Bmal1-S42A had little effect on the phases of these two gene oscillations (Fig. 6c), which suggesting that multiple mechanisms contribute to the diurnal oscillations of clock-controlled genes in the liver.

## Discussion

In the present study we demonstrate that insulin regulates Bmal1 transcriptional activity by reducing its nuclear accumulation via Akt-mediated Ser42 phosphorylation in the liver under physiological conditions (Fig. 6d). This revealed mechanism not only provides an explanation for the unusually long delay between the expression of Bmal1 and its regulated genes, but also sheds new

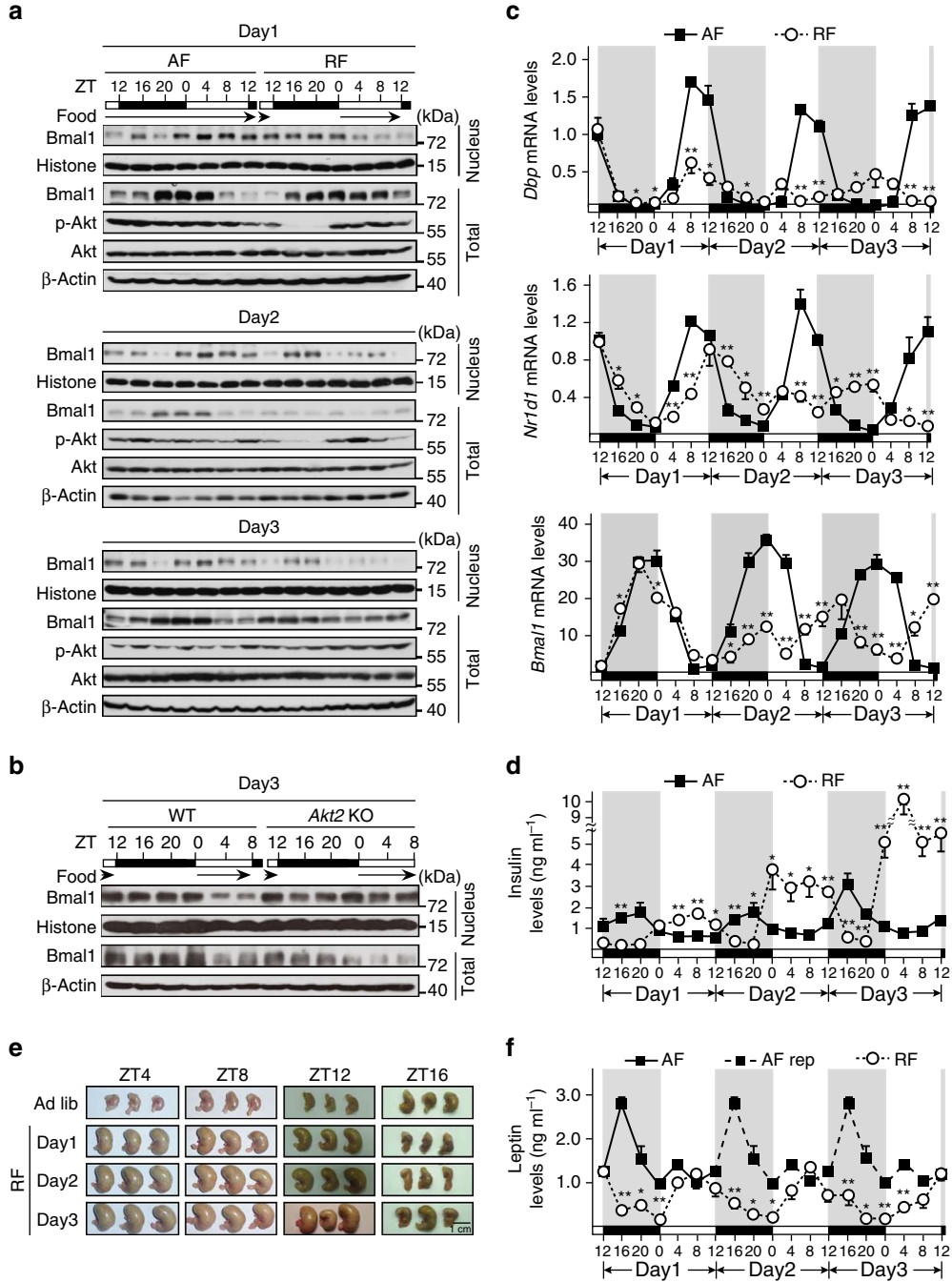

**Figure 5 | Temporal interplay between Bmal1 and insulin rhythms.** Mice were killed every 4 h over the first 72 h under either AF or RF initiated at Day1-ZT12, black arrows indicated food availability; plasma, total RNA, pooled liver total and nuclear protein extracts were prepared (**a**,**c**,**d**–**f**; $n = 3$). (**a**) Immunoblotting analysis of pooled nuclear and total Bmal1 protein levels. (**b**) Immunoblotting analysis of pooled nuclear and total Bmal1 protein levels in livers from WT or *Akt2* KO mice under RF during Day3 ($n = 3$). (**c**) Quantitative PCR analysis of *Dbp*, *Nr1d1* and *Bmal1* mRNA levels. (**d**) Measurement of plasma insulin levels by ELISA. (**e**) Pictures of mouse stomachs from AF and RF mice as indicated. (**f**) Measurement of plasma leptin levels. Leptin levels presented in Day2 and 3 were replicated from those of Day1 (AF rep, black squares and dash lines). Data are represented as mean ± s.e.m, statistical analyses were performed with a two-tailed unpaired Student's *t*-test, *$P < 0.05$, **$P < 0.01$ between groups at each time point.

light on understanding how food entrainment initiates the reset of the hepatic clock.

In contrast to established function of molecular clock in metabolic regulation, much less is known regarding the mechanism that adjusts peripheral rhythms to nutrient and hormonal cues. In line with several reports in which insulin has been shown to affect core clock gene expression in peripheral tissues[18–21,35], our study further reveals the underlying molecular

mechanism in this regard. Under *ad libitum* condition, master clock controls feeding behaviour to synchronize the phase and limit the signalling magnitude of insulin oscillation. Disrupting such synchronization, day feeding entrains it to an inverted phase and magnifies its output by stimulating insulin secretion due to food overconsumption that is driven by hungry anticipation. By overlapping with master clock-derived signalling wave, temporally enhanced signal from insulin

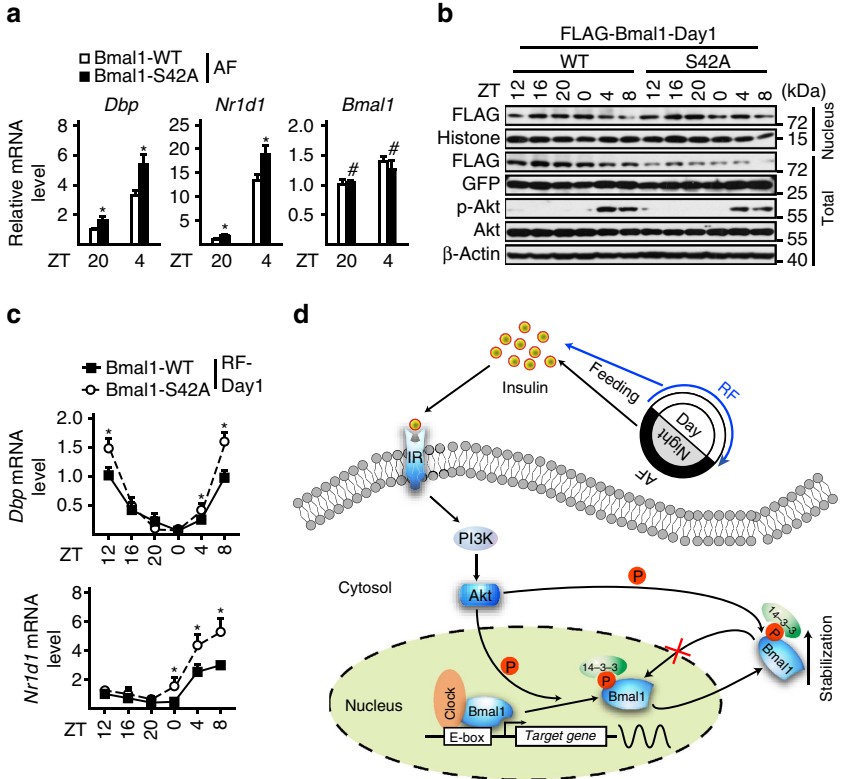

**Figure 6 | Effects of Bmal1-Ser42 phosphorylation on the hepatic clock.** (**a**) Quantitative PCR analysis of *Dbp* and *Nr1d1* mRNA levels in livers from AF mice tail-vein injected with FLAG-WT- or -S42A-Bmal1 adenoviruses and killed around ZT20 and ZT4 (*n* = 4). (**b**) Immunoblotting analysis of pooled nuclear and total FLAG-Bmal1 protein levels (*n* = 3) in livers from Day1-restrictedly fed (RF-Day1) mice tail-vein injected with FLAG-WT- or -S42A-Bmal1 adenoviruses. (**c**) Quantitative PCR analysis of hepatic *Dbp* and *Nr1d1* mRNA levels in the same samples as described in **b**. Data are represented as mean ± s.e.m, statistical analyses were performed with a two-tailed unpaired Student's *t*-test, *$P < 0.05$. (**d**) Hypothetical model showing the mechanism of insulin signal regulates Bmal1 in mouse liver. Postprandially, insulin inhibits Bmal1 transcriptional activity by promoting Akt-mediated Ser42 phosphorylation, which may not only dissociate Bmal1 protein from DNA to bind with 14-3-3, and followed by nuclear exportation, but also trap Bmal1 in the cytosol by masking its nuclear localization sequences.

initiates the adaption of the hepatic clock to the modulation of meal times.

Although our results reveal an essential role of insulin in the regulation of Bmal1 transcriptional activity by affecting its nuclear accumulation, the existence of other functional core clock machineries is indicated by the similar oscillatory patterns of hepatic Bmal1-target gene expression between *Akt2* knock-out and WT mice (Supplementary Fig. 3b) or between mice injected with Ad-Bmal1-S42A and Ad-Bmal1-WT viruses (Fig. 6c). The PER1/CRY1 inhibitory complex of BAML1/CLOCK is supposed to play a key role in this regard, since their nucleus-cytosol shuttling is insensitive to insulin (Fig. 1a). Moreover, the depletion of insulin may only delay but not block the reset of the hepatic clock, as other hormones (glucocorticoids) also affect *Bmal1* expression. Under circumstances without activating insulin signalling in the liver, for example STZ injection, the oscillation of nuclear Bmal1 accumulation should most likely follow the fluctuation of its total expression. Thus, RF should be still capable of resetting the former by reversing the latter through glucocorticoid signalling in the absence of insulin signalling activity. Supporting this notion, a previous report has shown that 10-day RF reverses the phase of plasma corticosterone levels, which leads to the reset of *Bmal1* and *Dbp* rhythms in STZ-treated mice[18]. Food anticipatory activity has long been considered to be the main force for resetting the peripheral clock[36]. Although our results do not address the origin of food anticipatory activity, they do point that refeeding hormonal signalling is a key link for adjusting molecular clock to

behavioural changes. Moreover, the existence of multiple feedback loops[5–8] and fasting/refeeding-regulated targets in molecular clock machinery[37–39] may contribute to the relatively long period (at least 1 week) for food entrainment to stabilize the peripheral clock.

Insulin promotes intracellular protein synthesis through AKT-mTOR-S6K signalling cascade. Considering a recent paper in which S6k has been shown to phosphorylate Ser42 residue on Bmal1 to keep it in the cytosol to facilitate protein synthesis[28], our findings suggest a scenario in which Akt and S6k temporally coordinate the inhibitory effect of insulin on Bmal1 transcritptional activity in the nucleus to ensure its function in translational machinery in the cytosol. Moreover, *Bmal1* deficiency has been shown to impair Akt activation by insulin[33], suggesting a close interplay between insulin signalling pathway and molecular core clock through the crosstalk between Akt and Bmal1. The importance of Akt phosphorylation in the regulation of Bmal1 protein stability has been indicated by the results that both *Akt2* deficiency and S42A mutation decrease total Bmal1 protein levels, which consists with a previous report that Bmal1 is destabilized via the phosphorylation of Ser17/Thr21 by Gsk3β, a kinase inhibited by Akt[40]. More interestingly, overexpression of Akt2-CA seems to stabilize Bmal1-S42A proteins (Figs 2d and 3g), which may be due to the sustained inhibition of Gsk3β by constitutively activation of Akt2.

The surprisingly high levels of plasma insulin in RF mice during Day2 and 3 suggest that β-islet cells suffer a great stress during food entrainment, which would cause the

apoptosis/dysfunction of β-cells[41]. Actually, a recent paper has shown that a long term of RF induces diabetes, fatty liver and obesity in mice[42]. Thus, our results provide a new clue to understand the link between perturbation of the circadian rhythm system and the development of metabolic syndrome.

## Methods

**Animals and experimental design.** The $Akt2^{-/-}$ mouse line was generously provided by Dr Zhongzhou Yang (Laboratory of Heart and Disease Model, Model Animal Research Institute, Nanjing University, Nanjing, China); the *Alb-Cre* mouse line was generously provided by Dr Yong Liu (Institute for Nutritional Sciences, China); C57BL/6 WT mice were purchased from Shanghai Laboratory Animal Center, China; and the $Bmal1^{fl/fl}$ mouse line was purchased from Jackson Laboratories. The $Bmal1^{fl/fl}$ mouse line was crossed with *Alb-Cre* line to generate $Bmal1$ liver-specific knock-out mice. The 8- to 12-week old male mice were used in all the experiments and housed in the animal facility at Shanghai Institutes for Biological Sciences (SIBS). Mice were maintained on a 12-h light/12-h dark cycle for at least 2 weeks before study, had free access to water and regular diet and were randomly allocated into treatment groups. The number and strain of mice used in each experiment were specified in the corresponding figure legend. For RF experiments, food was removed 15 min before darkness and provided 15 min after light. After killing mice, blood and liver samples were collected. Plasma insulin levels were measured by ELISA kits (Millipore). The investigators were not blinded to allocation during experiments and outcome assessment. All animal care and use procedures were ethically approved by the Animal Care and Use Committee of Shanghai Institutes for Biological Sciences.

**Plasmids and adenoviruses.** Plasmid expressing mouse Bmal1, Bmal1 (S42A), Bmal1 (S422A), Bmal1 (S513A), Bmal1 (S422A/S513A), Clock and Akt2 were constructed from mouse cDNA with specific primers as shown in Supplementary Table 1. *Per1-Luc* has been described previously[43]. pcDNA-Myr-FLAG-Akt2 (Akt2-CA) was constructed according to previously described[44]. Ad-FLAG-tagged Akt2-CA, Ad-FLAG-tagged WT or S42A mutant Bmal1 adenoviruses were constructed by using AdEasy system (Agilent Technologies). GFP adenovirus has been described previously[45]. In all, $1 \times 10^8$ plaque-forming units (p.f.u.) of Bmal1-WT or $1.2 \times 10^8$ p.f.u. of Bmal1-S42A adenoviruses were delivered to male C57BL/6J mice by tail-vein injection. Mice were killed 7 days after injection.

**Cell culture.** HEK 293T and HepG2 cells were purchased from ATCC, cultured in DMEM supplemented with 10% fetal bovine serum, 1% penicillin/streptomycin and 1% l-glutamine and checked for mycoplasma by PCR or ELISA. For luciferase reporter assay, HEK 293T cells were transfected using 50 ng of *Per1-Luc* and 20 ng of β-gal, together with 100 ng of Bmal1 and Clock plasmids per well. Co-transfections were performed with a constant amount of DNA by adding 100 ng of the empty vector pcDNA3 (Invitrogen) and luciferase activity was measured. Mouse primary hepatocytes were prepared, cultured and infected with adenoviruses. Briefly, the hepatocytes were isolated by using collagenase (Sigma) and plated on dishes coated with rat tail tendon collagen (Sigma) according to manufacturers' instructions. Before experiments, all the cells were synchronized by 12 h serum starvation. For experiments with LY-294002 (Sigma, LY) or LMB (Cell Signaling Technology, LMB), hepatocytes were pre-treated with LY (10 μM) or LMB (20 ng ml$^{-1}$) for 2 or 5 h, before incubation with insulin (50 nM) for indicated times.

**Luciferase activity assay.** HEK 293T cells were cultured in 24-well plates, collected after 24 h of transfection and then lysed by adding 150 μl per well of the Gly-gly lysis buffer (25 mM Gly-gly, pH 7.8; 15 mM MgSO$_4$; 4 mM EGTA, pH 7.8; 1 mM DTT and 1% V/V Triton X-100). The plates were shook gently for 20 min by transference decoloring shaker, 50 μl per well of lysate was transferred to a well of a black 96-well plate, and then mixed with 50 μl of the luciferase assay mixture (20 mM Gly-gly, pH 7.8; 12 mM MgSO$_4$; 3 mM EGTA, pH 7.8; 0.2 mM potassium phosphate, pH 7.8; 2 mM ATP; 1.5 mM DTT and 1.25 mg ml$^{-1}$ firefly luciferin). Fluorescence intensity was measured by using Luninoscan Ascent (Thermo). For β-gal assay to normalize the luciferase assay results, another 50 μl per well of lysate was taken to be mixed with 50 μl per well of β-gal solution, incubated at room temperature until colour developed and then read at A420 on a plate reader (Epoch Microplate Spectrophotometer, BioTek).

**Total RNA isolation and quantitative PCR assay.** Total RNAs from whole livers or primary cultured hepatocytes were extracted with Trizol (Invitrogen) and reverse-transcribed into cDNAs by prime script RT regent kit with gDNA eraser (Takara) according to the manufactures' instructions. Relative mRNA levels were determined by SYBR-green PCR kit with ABIPRISM 7900HT Sequence detector (Perkin Elmer) with specific primers as shown in Supplementary Table 2. Ribosomal L32 mRNA levels were used as internal control.

**Immunoblotting assay.** For immunoblot analysis, cultured cells or collected liver tissues were lysed in RIPA buffer containing proteinase inhibitors (PMSF, Cocktail and Phosphatase Inhibitors), followed by centrifuging at 12,000 r.p.m. for 15 min at 4 °C. Supernatants were collected and protein concentrations were determined by using BCA protein assay kit (Pierce, 23225) according to the manufactures' instructions. Identical amounts of proteins were subjected to SDS-PAGE electrophoresis and transferred to methanol-activated PVDF membranes (Millipore), which were blocked with 5% milk for 1 h at room temperature, incubated overnight with corresponding primary antibodies at 4 °C and then with secondary antibodies for 1 h at room temperature. Finally, the protein bands were developed by using ECL western blotting substrate (Thermo Pierce) and visualized by western film processor. Primary antibodies used in the experiments were as followed: Bmal1 (Abcam, catalogue No. ab3350, diluted 1:1,000); Clock (Abcam, catalogue No. ab3157, diluted 1:1,000); Histone H3 (Cell Signaling Technology, catalogue No. #9715, diluted 1:4,000); pSer473-Akt (Cell Signaling Technology, catalogue No. #9271, diluted 1:1,000); Akt (Cell Signaling Technology, catalogue No. #9272, diluted 1:1,000); FLAG-HRP (Sigma, M2, catalogue No. A8592, diluted 1:4,000 for immunoblotting Flag-tagged proteins); 14-3-3β (Santa Cruz Biotech, K-19, catalogue No. sc-629, diluted 1:3,000); Phospho-Akt Substrate (Cell Signaling Technology, 110B7E, catalogue No. #9614, diluted 1:1,000); β-Actin (Cell Signaling Technology, catalogue No. #4967, diluted 1:5,000) and α-Tubulin (Cell Signaling Technology, catalogue No. #2144, diluted 1:5,000). Uncropped blots are shown in Supplementary Fig. 5.

**Immunoprecipitation assay.** For immunoprecipitation analysis, liver tissues or cultured primary hepatocytes were lysed in the immunoprecipitation buffer (IP, 25 mM Tris-HCl, pH 7.4, 150 mM NaCl, 1 mM EDTA, 0.1% NP-40, 1% Triton X-100, 10% glycerol and proteinase inhibitors), followed by centrifuging at 12,000 r.p.m. for 15 min at 4 °C. Supernatants were collected and their protein concentrations were adjusted to 1 μg μl$^{-1}$ with the BCA protein assay results. In all, 400–600 μl of each remaining homogenate was incubated with indicated antibodies together with IP-buffer-washed protein A agarose beads (16–156, Merck Millipore) overnight or with anti-FLAG M2 Magnetic beads (Sigma) for 2 h at 4 °C. The immunoprecipitation beads were then washed with IP buffer for six times, and subjected to immunoblotting analysis.

***In vitro* kinase assay.** For *in vitro* kinase assay, FLAG-tagged WT or S42A Bmal1 proteins were immunoprecipitated from HEK 293T cells transfected with the corresponding plasmids. The proteins were incubated with 20 ng μl$^{-1}$ purified GST-Akt (Biovision) in the kinase buffer (25 mM Tris-HCl, pH 7.5, 5 mM β-Glyverophosphate, 2 mM DTT, 0.1 mM Na$_3$VO$_4$, 10 mM MgCl$_2$, 200 μM ATP) at 30 °C for 30 min. The reaction was stopped by boiling samples in SDS-PAGE loading buffer and then separated by SDS-PAGE. Phosphorylation signals were detected by using anti-phospho-Ser42-Bmal1 antibody (1:500).

**Chromatin immunoprecipitation.** Liver tissues or primary hepatocytes were cross-linked with 1% formaldehyde. Bmal1 or FLAG antibodies were used for immunoprecipitations along with rabbit IgG for negative controls. After removing crosslinks, DNA was extracted by using phenol–chloroform and precipitated with ethanol. Target promoters were analysed by using quantified SYBR-green real-time PCR and normalized to input chromatin signals with primers as shown in Supplementary Table 3.

**Cell/tissue fractionation.** Hepatocytes or liver tissues were washed two times with ice-cold PBS and pelleted at 3,000 r.p.m. at 4 °C. Pellets were resuspended in buffer containing 0.3% NP-40 for cells and 0.3% Triton X-100 for liver tissues in PBS with protease inhibitors. Cells/tissues were lysed with a dounce homogenizer and centrifuged at 4,200 r.p.m. for 10 min at 4 °C. Cytosolic supernatants were collected. Nuclear pellets were washed and resuspended in 200 μl nuclear extraction buffer (20 mM Tris, pH 7.4, 450 mM NaCl, 1% NP-40, 1 mM DTT, with protease inhibitors). Samples were sonicated and centrifuged at 13,000 r.p.m. for 20 min before collection of nuclear supernatants.

**Immunostaining assay.** Primary cultured hepatocytes were pre-synchronized by 12 h serum starvation. After treatment, hepatocyte cells were fixed in 4% paraformaldehyde for 20 min, permeabilized with 0.3% triton X-100 and incubated with an anti-Bmal1 (1:1,000 dilution) or anti-FLAG antibody (1:1,000 dilution) for endogenous Bmal1 or FLAG-Bmal1 detection, respectively. Slides were washed and mounted with coverslips by using Vectashield mounting media containing 4, 6-diamidino-2-phenylindole.

**Mass spectrometry.** Mass spectrometry studies were performed to identify the Bmal1 phosphorylation residues. Briefly, primary hepatocytes were infected by the FLAG-tagged Bmal1 adenovirus and then treated with or without insulin (50 nM) for 30 min. Immunoprecipitates of FLAG-Bmal1 proteins were separated by 10% gradient SDS-PAGE gels and stained with Coomassie Brilliant Blue for 15–20 min

followed by de-staining at room temperature for 2 h. Entire lanes were cut into pieces and sent to Shanghai Applied Protein Technology Co. Ltd (China) for liquid chromatography-mass spectrometry (LC-MS/MS) analysis with the equipment of LTQ VELOS (Thermo Finnigan, San Jose, CA). The principle of peptide spectrum match was applied to characterize the phosphorylation residue on Bmal1.

**Statistical analyses.** Results were represented as mean ± s.e.m. Statistical tests were selected based on appropriate assumptions with respect to data distribution and variance characteristics. The comparisons of two groups of mice or different primary hepatocytes preparations were carried out using two-tailed unpaired Student's *t*-test by Graphpad Prism5 (Graphpad Software, San Diego, CA, USA). Differences were considered statistically significant at $P < 0.05$. No statistical methods were used to predetermine sample size. All experiments were performed on at least two independent occasions.

**Data availability.** The authors declare that the data supporting the findings of this study are available within the article and its Supplementary Information files.

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

## Acknowledgements

We thank Zhixue Liu (SIBS, CAS, China) for the gift of GST-Akt recombinant protein, Zhongzhou Yang (Model Animal Research Center of Nanjing University, China) for the gift of Akt2 knock-out mice and Yong Liu (Institute for Nutritional Sciences, China) for the gift of Alb-Cre mice. This work was supported by grants from the National Natural Science Foundation of China (81390351), National Basic Research 973 Program (2014CB910500), NSFC (31471123 and 31222028), NBR973 (2012CB524900) and the Chinese Academy of Sciences (XDA12040306).

## Author contributions

F.D., X.S. and Y.L. designed and conceived the study. F.D. and X.S. performed most of the experiments and data analysis. X.M. performed experiments to identify phosphorylation of S42 on Bmal1. D.Z. assisted with food entrainment experiments. Y.W. assisted with adenovirus preparation and mouse line maintenance. R.W., Y.C. and Q.X., helped perform experiments during revision. Y.L. helped interpret data and with input from all other authors, developed the final version of the manuscript.

## Additional information

**Competing financial interests:** The authors declare no competing financial interests.

