## [Peer Review File · Nature Communications]

Reviewers' comments:

Reviewer #1

Expert in circadian biology

(Remarks to the Author):

Dang et al. describe in this article how feeding may affect clock phase in the liver. They show that after feeding insulin promotes the phosphorylation of Ser42 of BMAL1 via AKT signaling, an alternative way to the described TOR mediated mechanism. Phosphorylation at Ser42 induces BMAL1 dissociation from DNA and nuclear export via 14-3-3 protein resulting in reduction of BMAL1 mediated transcription. This mechanism on BMAL1 regulation has not been described before and may be of interest to others in the fields of circadian rhythms and obesity research. Although of interest the paper has a few shortcomings:

1) Most of the presented protein data is not quantified and no statistical analysis of these data are provided.

2) The issue of interest is the distribution of BMAL1 in the nucleus and the cytosol. The authors provide this data in several figures but in others only nuclear extracts are compared to total extracts (e.g. Fig. 1c,d, 2g, 4a). Is there a rationale behind this or did the cytosolic extracts simply not look good enough?

3) A discussion is missing completely. The authors may discuss their findings and put them in relation to the recent papers by Mukherji et al., 2015, PNAS E6683-E6690 and E6691-E6698. They describe the effects of shifted feeding, without providing the upstream mechanism described here. How do the authors envisage the regulation of Ser42 phosphorylation of BMAL1 in vivo? Is there a competition between the S6K and AKT? What nutritional state would favor S6K over AKT or vice versa?

Minor points:

1) Scale bars are missing in the Immunohistochemistry data (Fig. 1b).

2) The writing should be improved. There are passages that are hard to understand and in some cases may even lead to misunderstandings. This is a very important issue!

3) Figures are mislabeled and do not conform to the citation in the text, which leads to confusion when reading the paper. Please correct this and carefully adapt the text or figures.

4) Figure S2a is not well described and therefore is not easy to understand for a non specialist in MS.

5) The methods are in many cases rudimentary and should be more detailed (e.g. immunohistochemistry, Mass Spectrometry).

Reviewer #2

Expert in circadian entrainment

(Remarks to the Author):

Liu and colleagues contribute to disentangle what are the molecular mechanisms by which the feeding schedule is able to reset peripheral clocks. Hereto insulin was taken as a prototype of the feeding signal, and one of its main downstream signaling pathways, the one mediated by the

serine/threonine kinase Akt was shown to phosphorylate BMAL1 in a conserved serine residue, relating this with its subsequent association with 14-3-3 proteins, nuclear export and decrease in its binding to some of its DNA target sites. The authors also show that when the feeding schedule is abruptly changed, the change in the levels of nuclear BMAL1 is an early event, preceding the changes in Bmal1 gene transcription itself, total BMAL1, and mRNA levels of downstream genes, suggesting that the signaling cascade mediated by insulin-AKT-BMAL1 may have a key role in the clock resetting by food.

Comments:

In general, the paper needs a careful revision of spelling, an improvement of the Methods section, as well as to include a careful discussion of the results.

For example

a) The authors determined that an insulin injection to fasted animals "led to significant attenuation of BMAL1 occupancy on the promoters of its target genes (e.g. Dbp and Nr1d1, fig. 1e) and subsequent reduction of their transcription (fig 1f)". Previous studies have shown that insulin indeed increases transcription of two core clock genes: Per1 and Per2 (Yamajuku et al. 2012), which seems incompatible with their observation, do the authors have an explanation?

b) Page 4, last four lines: The authors affirm that: "Inhibition of Dbp expression by insulin was abolished in Bmal1 null primary hepatocytes (fig. 1g), suggesting such insulin effect is mediated by BMAL1". The results should be discussed in light of a preceding paper that demonstrates Bmal1 deficiency disrupting the induction of phospho-AKT by insulin (Zhang et al., JBC 2014), thus in this experiment they should have introduced an adenovirus with a constitutive active form of AKT, in order to see the specific contribution of BMAL without the confusion of upstream defects in the insulin signaling pathway.

c) Page 6: "This notion was further supported by the in vitro results that S42A mutation unstabilized BMAL1 in HEK 293T cells (sup. fig. 2e)". There seems to be variability between some of the experiments, or there is an important cellular context-dependent effect, that should be addressed. In this case, at time 0 (before treatment sup.fig.2e) the levels of BMALmutant in HEK293T as judge by the flag signal are reduced when compared with the WT at the same time 0. But the basal levels of whole BMAL1 in primary hepatocytes infected with the mutant are clearly higher before any treatment than those hepatocytes infected with the WT (Fig 2c). These same differences before treatment in BMAL whole levels also occurs in AKT2 KO experiments, where in in vivo experiments BMAL1 levels are lower in the KO than in the WT, but the primary hepatocytes of the AKT2 KO mice have previous to treatment BMAL levels (as judge by the signal obtained with the flag-BMAL1 antibody) comparable to those of the WT. So at least with regard to total BMAL stability standardization is needed.

d) Page 7-8: "As a result, BMAL1 mutants containing S42A induced much higher Per1-Luc activity than that of wild-type BMAL1 (20-fold over 8 fold) with co-expression of CLOCK in HEK293T cells and neither S422A nor S513A mutation exhibited similar effects (fig. 3g), suggesting the potential role of Ser2 phosphorylation in suppressing basal transcriptional activity of BMAL1". Why did the authors include the other mutants here? Why, given that they already had the constructs, were the mutants not used in the initial experiments aimed to disentangle the key residue for AKT-phosphorylation (e.g. 2c-and beyond, sup. fig2)?

e) Regarding the assertion in page 8: "these results suggest that phosphorylation of Ser 42 by insulin triggers BMAL1 to dissociate from DNA..."; it is recommended to write instead: "may trigger BMAL1 to dissociate", because the experiments presented are not sufficient to demonstrate that the less or the more BMAL1 on the chromatin is due to the variation in BMAL1 levels within

the nucleus or to a change in its affinity for target sequences.

f) Page 8: "However, the role of insulin signal in food entrainment resetting hepatic clock is still unknown." Minimally here the studies of restricted feeding and circadian gene expression done in streptozotocin-treated or other animal models of diabetes should be cited (Oishi et al., 2004; Hoffman et al., 2013; Tseng et al., 2015).

g) Page 9: The authors mention a reduction in leptin rhythm amplitude. Why did they measure leptin? Instead, or in addition to, they should have measured glucagon, which is known to affect Bmal1 (Sun et al., JBC 2014). And discuss it.

h) Page 9: "Taken together, our results revealed an essential role of insulin in the regulation of BMAL1 transcriptional activity by determining its nuclear localization via AKT-mediated S42 phosphorylation in the liver under physiological conditions (sup. fig 4f), which plays an initiating and sustaining function in hepatic clock reset by food entrainment". This notion is not fully supported by studies in diabetic rats (Oishi et al., 2004). The authors should discuss the discrepancies. Additionally, it is known that BMAL also affects the capacity of AKT for being activated by insulin (Zhang et al., JBC 2014). It would be also interesting to discuss it.

Methods:

Methods should be more detailed and carefully written before publication. Some examples for improvements are:

i) Animals and Experimental Design. The Akt2^{-/-} mouse line was from Z. Yang (to give the complete name and institution). The Alb-Cre mouse line was from Y Liu; and the Bmal1^{fl/fl} mouse line was purchased from Jackson Laboratories: at which experiments were these animals included? Specify.

j) Cell culture. Given the importance of culture conditions for circadian clock gene expression (Yamajuku et al. Sci Rep 2012), the authors have to be more explicit in their methods.

k) Page 12, Immunoblot: in which experiment was the HSP90 antibody used? It is needed the catalogue number of all the antibodies used, for reproducibility purposes. Page 13, Mass spectrometry: absolutely scarce, how did they prepared the samples, which mass spectrometry method and/or equipment was utilized? Which criteria did they use to characterize the phosphorylation sites?

Immunostaining. Important, specify after how many hours after serum synchronization were the experiments done.

l) The experiments done in vivo, with AKT2 KO animals (Figures 2g and 4b) would be suitable complemented (and the conclusions strengthened) using three animals injected in the tail-vein with Ad-FLAG-S42A-BMAL1 (the authors already have the adenoviral construction, as indicated in Fig 3H legend).

m) Fig. 3h. There are required control no-refed groups, in order to compare the changes in BMAL phosphorylation in response to a physiological challenge. Otherwise, how could be explained the abrupt change in DBP and NR1D1 proteins at just 30 minutes after re-feeding?

Figures/Results:

n) Fig 1d. The subtitle "fasted" should be instead "time post-injection".

o) Fig 2c. Why the authors did not use an antibody to p-AKT-Substrate as in 2a and 2b, in order to

know if the other two putative AKT phosphorylation sites in BMAL1 protein are also phosphorylated by insulin stimulation?

p) Fig. 2f. In this, as in Sup. fig. 2c it is suggested a Coomassie staining, to ensure the amount of protein is similar, and to ensure that it didn't have a proteolytic event in the protocol process. To include in Methods sf9 cells.

q) Fig. 2g. Do AKT2 KO animals have a food intake circadian rhythm similar to WT? It is important to indicate that in the text. In order to confirm the hypothesis that akt2 signaling is essential for nuclear BMAL1 circadian rhythm, it would be useful to compare the results with those of liver-specific constitutive-active AKT2 mutant mice.

r) Fig. 3f. Why are the input signals so weak?

s) Fig. 3g. I do not understand why Per1 transcription is not affected when the S422A and S513 variants are added conjointly with the S42A mutant (black bars)? Were they used in equal quantities? One could expect a decrease in the transcription rate due to competition; with a consequent bar length between the one of WT+CLOCK and S42A+CLOCK.

t) Do AKT2 KO mice have no circadian rhythms in the clock-output genes?

u) Fig. Suppl.1: Histone and beta-actin groups are interchanged regarding their nuclear or whole localization.

v) Fig. Suppl.2a: Indicate the m/z where is located the phosphorylation related residue, and indicate it in the diagram. The authors should show a similar figure for peptides containing the other two possible phosphorylation sites of BMAL1.

w) Fig. Suppl. 2c: Why is the Flag signal so weak in the WT?

x) Fig Suppl. 3f: Why is insulin-degrading-enzyme (IDE) here?

Reviewer #3

Expert in insulin signalling

(Remarks to the Author):

Dang et al report that insulin promotes Akt-dependent phosphorylation of S42 in BMAL1, thereby prompting its association with 14-3-3 proteins and redistribution from the nuclear to the cytoplasmic compartment and disrupting its effects on gene expression, including its ability to function as a partner with CLOCK in the regulation of diurnal rhythms in the liver. The authors propose that insulin/Akt signaling to BMAL1 S42 accounts for the ability to reset the biological clock in the liver in response to alterations in feeding patterns.

The authors provide clear evidence indicating that BMAL1 is a target of insulin/Akt signaling, and that Akt phosphorylation of Ser42 plays an important role in the recruitment of 14-3-3 proteins and nuclear exclusion of BMAL1 in hepatocytes. This is a novel, and important result, and it is reasonable to speculate that these effects will have an important impact on clock function in the liver.

At the same time, it remains possible that other mechanisms also may contribute to re-entraining the clock in the liver when the timing of food intake is altered, and additional studies evaluating the relative importance of this specific mechanism in mediating the effect of insulin and feeding on

the expression of known BMAL1 downstream targets and function of the clock would further strengthen the manuscript and enhance its potential impact.

Major Concerns:

1. Studies in Akt2 KO mice support the concept that Akt 2 plays an important role in promoting nuclear exclusion of BMAL1, but they do not demonstrate that S42 is critical for this effect. Additional studies are needed to show that S42 is required to mediate effects of Akt on BMAL1 nuclear/cytoplasmic trafficking, and regulation of clock function. One experiment would be to express BMAL1-GFP fusion proteins with/without mutation of S24 and show that replacement of S24 with alanine disrupts the ability of insulin to promote nuclear exclusion of BMAL-GFP in an Akt-dependent fashion (e.g. with/without Akt inhibition). Co-transfection studies with constitutively active Akt also would demonstrate direct S42-dependnet effects on BMAL1 translocation.
2. Similarly, studies are needed to determine whether S42 phosphorylation is required to mediate effects of insulin on the expression of downstream targets of BMAL1 (e.g. Dbp). If this is the case, overexpression of S42A BMAL1 would be expected to disrupt the ability of insulin (and/or Akt) to suppress the expression BMAL1 target genes. Similar studies can be performed utilizing promoter assays to show whether BMAL1 protein and cis-acting BMAL1 sites are required to mediate effects of insulin/Akt on promoter activity, and whether overexpression of S42A BMAL1 blocks these effects of insulin/Akt.
3. In vivo studies addressing the role of S42 phosphorylation in the regulation of clock function in the liver also would strengthen the study. Short of creating an S42A knock-in mouse, adenoviral expression of S42A BMAL1 in the liver would be expected to result in constitutively nuclear BMAL1, and disrupt both the regulation of clock in the liver, and the ability to reset the hepatic clock altering the timing of feeding. If these experiments succeed, they will provide clear evidence for S42 phosphorylation in the regulation of hepatic clock activity. If they do not show this result, they will suggest that other mechanisms are sufficient to maintain effects of feeding time on clock activity in the liver.

Other Specific Comments:

1. The authors utilize a site-specific antibody to measure phosphorylation of serine 42 in BMAL1. Since this appears to be a new reagent that has not been previously reported, please describe how this antibody was made, and provide data demonstrating its specificity, including full length western blots and competitive inhibition studies with phospho- and non-phosphopeptides.
2. p. 3, last sentence. The authors state several times that insulin "stimulates" translocation of BMAL1 from the nucleus to the cytoplasmic compartments. Since S42 is located within a nuclear localization signal, a more likely scenario is that phosphorylation of S42 masks this NLS, thereby preventing translocation from the cytoplasmic compartment into the nucleus, thereby trapping it in the extranuclear space. The text and summary figure should be revised to include this possibility.
3. p. 4, line 6. Remove the "s" at the end of "accumulations".
4. p. 4, last line. Suggest replacing "determine" with "limit".
6. p. 5, line 4. Insert "of" after "exposure".
7. p. 6 and Fig 2g.
8. The authors note that S42A BMAL1 is more efficiently recruited to the Dbp and Nrfd1 promoters, and activate the Per1 promoter, and suggest that this reflects improved DNA binding. However, It seems more likely that this may simply reflect differences in the level of BMAL1 protein (S42A >> WT) in the nucleus, and not differences in DNA binding efficiency. To show that there are differences in DNA binding efficiency, additional studies would be needed to assess binding activity

under conditions where WT and S42A protein levels are comparable - e.g. in gel shift assays where levels of WT and S42 BMAL1 can be controlled. (Note: if differences are observed, 14-3-3 proteins may contribute to this effect.). Without this data, the authors need to adjust the text to allow for the possibility/likelihood that nuclear exclusion of WT (but not S42A) BMAL1 contributes to differences in transcriptional activity.

10. fig 3 f. The FLAG western blot for protein input is weak. Can the authors provide a more convincing western?

11. sup fig 3. What is "AF rep"?

12. sup fig 3f. What is the significance of IDE? This is not mentioned in the text. Please explain, or delete.

According to reviewers' comments and the regular format of the papers published on *Nature Communications*, we have made a major revision of our manuscript, in which we have added Introduction and Discussion sections and reorganized figures. For reviewers' convenience, we provided corresponding figure numbers used in the last submitted version and current manuscript in below responses.

Reviewer #1

1) Most of the presented protein data is not quantified and no statistical analysis of these data are provided.

Answer: We have quantified and statistically analyzed most of the protein data in this revised manuscript (e.g. Fig. 1d, 2b, S2c, S2f, S2h, S3a, S4b, and S4c in the current manuscript, corresponding to Fig. 1d, 2b, 2c, 2d, 2e, 2g, 4a, and 4b in the last submitted version).

2) The issue of interest is the distribution of BMAL1 in the nucleus and the cytosol. The authors provide this data in several figures but in others only nuclear extracts are compared to total extracts (e.g. Fig. 1c, d, 2g, 4a). Is there a rationale behind this or did the cytosolic extracts simply not look good enough?

Answer: To keep nuclear extracts from cytosolic contamination, we used stringent conditions to ensure the lysis of all the cells, especially those in liver samples, which made cytosolic extracts contaminated from nuclear parts. As we mainly focused on studying the nuclear accumulation of BMAL1, we didn't provide cytosolic BMAL1 data in all figures in the last manuscript. In the present version of manuscript, we provided the data of cytosolic BMAL1 protein levels in the corresponding liver samples (Fig. 1c, 1d and 3a in the current manuscript, corresponding to Fig. 1c, 1d and 2g in the last submitted version).

3) A discussion is missing completely. The authors may discuss their findings and put them in relation to the recent papers by Mukherji et al., 2015, PNAS E6683-E6690 and E6691-E6698. They describe the effects of shifted feeding, without providing the upstream mechanism described here.

How do the authors envisage the regulation of Ser42 phosphorylation of BMAL1 in vivo? Is there a competition between the S6K and AKT? What nutritional state would favor S6K over AKT or vice versa?

Answer: We have added the Discussion section in the revised manuscript and discussed the issues proposed by Reviewer1 (page 15, line 11; page 17, line 15). Briefly, the data about the effects of shifted feeding on the expression profiles of core clock members (e.g. *Bmal1* and *Rev-erba*) are consistent between the two studies by Mukherji et al., and us. In addition to the involvement of PPAR α and Glucagon receptor in RF resetting peripheral clock in their papers, our results provide new mechanism to show that insulin play an important role to initiate the phase shift of hepatic clock by affecting nuclear accumulation of BMAL1 proteins.

S6K has been shown to be activated by insulin-AKT-mTOR signaling cascade. It

has been shown that insulin induces AKT activity within 5 minutes, whereas, S6K activity much later^{1,2}. Thus, it's very likely that there's no competition between S6K and AKT for the regulation of Ser42 phosphorylation of BMAL1, instead, these two kinases temporally coordinate the inhibition of insulin on nuclear accumulation of BMAL1 proteins via the phosphorylation of its Ser42 residue, which is initiated by AKT and maintained by S6K. We discussed this issue in the revised manuscript (page 8, line 18, page 16, line 19).

4) Scale bars are missing in the Immunohistochemistry data (Fig. 1b).

Answer: We have added scale bars in Fig. 1b in the current version.

5) The writing should be improved. There are passages that are hard to understand and in some cases may even lead to misunderstandings. This is a very important issue!

Answer: We are sorry for our mistakes. We tried our best to improve the writing and correct the mistakes, and had our manuscript checked by a professional English editor.

6) Figures are mislabeled and do not conform to the citation in the text, which leads to confusion when reading the paper. Please correct this and carefully adapt the text or figures.

Answer: We are sorry for our mistakes. We have correct this mistake and adapted the text and figures in the present manuscript.

7) Figure S2a is not well described and therefore is not easy to understand for a non specialist in MS.

Answer: We have described this experiment in details in the Method section (page 22 line 21) and annotated the phosphorylation site in Supplementary Fig. 2a (up) in the revised manuscript.

8) The methods are in many cases rudimentary and should be more detailed (e.g. immunohistochemistry, Mass Spectrometry)

Answer: We have detailed the methods in our revised version, and thank for Reviewer1's constructive suggestion (page 18-23).

Reviewer #2:

a) The authors determined that an insulin injection to fasted animals "led to significant attenuation of BMAL1 occupancy on the promoters of its target genes (e.g. Dbp and Nr1d1, fig. 1e) and subsequent reduction of their transcription (fig 1f)". Previous studies have shown that insulin indeed increases transcription of two core clock genes: Per1 and Per2 (Yamajuku et al. 2012), which seems incompatible with their observation, do the authors have an explanation?

Answer: Although it seems that our data conflict with the conclusion of the work by Yamajuku et al., there's actually no discrepancy between these two studies. There's an interesting phenomenon of BMAL1 regulating *Per1* and *Per2* expression in vivo: *Bmal1* knockout has been reported to increase the mRNA levels of *Per1* and *Per2* in mouse liver, in contrast, decrease *Dbp* and *Nr1d1* expression^{3,4}. As insulin suppressed BMAL1 transcriptional activity, we expected to observe similar events in mice with insulin injection. Indeed, our data showed that intraperitoneal injection of insulin enhances hepatic expression of *Per1* and *Per2* in vivo (please see below figure). Thus, our results are consistent with other published literatures.

Quantitative PCR analysis of mRNA levels of *Per1* and *Per2* in insulin-injected mice.[i.p insulin (2U/Kg) at ZT4 and sacrificed at ZT6, n=3, ** p<0.01]

b) Page 4, last four lines: The authors affirm that: "Inhibition of *Dbp* expression by insulin was abolished in *Bmal1* null primary hepatocytes (fig. 1g), suggesting such insulin effect is mediated by BMAL1". The results should be discussed in light of a preceding paper that demonstrates *Bmal1* deficiency disrupting the induction of phospho-AKT by insulin (Zhang et al., JBC 2014), thus in this experiment they should have introduced an adenovirus with a constitutive active form of AKT, in order to see the specific contribution of BMAL without the confusion of upstream defects in the insulin signaling pathway.

Answer: We have performed the experiments according to Reviewer2's suggestion and the data showed that the overexpression of a constitutive active AKT2 (AKT2-CA) significantly reduces *Dbp* expression in WT primary hepatocytes, but not in *Bmal1* null hepatocytes (Supplementary Fig. 3i), which supports our hypothesis.

c) Page 6: "This notion was further supported by the in vitro results that S42A mutation unstabilized BMAL1 in HEK 293T cells (sup. fig. 2e)". There seems to be variability between some of the experiments, or there is an important cellular context-dependent effect, that should be addressed. In this case, at time 0 (before treatment sup.fig.2e) the levels of BMALmutant in HEK293T as judge by the flag signal are reduced when compared with the WT at the same time 0. But the basal levels of whole BMAL1 in primary hepatocytes infected with the mutant are clearly higher before any treatment than those hepatocytes infected with the WT (Fig 2c). These same differences before treatment in BMAL whole levels also occurs in AKT2 KO experiments, where in in vivo experiments BMAL1 levels are lower in the KO

than in the WT, but the primary hepatocytes of the AKT2 KO mice have previous to treatment BMAL levels (as judge by the signal obtained with the flag-BMAL1 antibody) comparable to those of the WT. So at least with regard to total BMAL stability standardization is needed.

Answer: Since BMAL1-S42A protein is unstable, we intentionally infected cells with more adenoviruses expressing BMAL1-S42A than WT control to ensure that the abolishment of S42 phosphorylation is not due to less total BMAL1 protein levels. As Reviewer2 suggested, we normalized BMAL1-WT or S42A protein levels with those of GFP that is constructed in the same virus with BMAL1, and showed that the relative protein levels of BMAL1-S42A are lower, compared with WT control (Fig. 2b and Supplementary Fig. 2c and 2h) in the current manuscript, corresponding to Fig. 2b, 2c and 2e in the last version).

d) Page 7-8: "As a result, BMAL1 mutants containing S42A induced much higher Per1-Luc activity than that of wild-type BMAL1 (20-fold over 8 fold) with co-expression of CLOCK in HEK293T cells and neither S422A nor S513A mutation exhibited similar effects (fig. 3g), suggesting the potential role of Ser2 phosphorylation in suppressing basal transcriptional activity of BMAL1". Why did the authors include the other mutants here? Why, given that they already had the constructs, were the mutants not used in the initial experiments aimed to disentangle the key residue for AKT-phosphorylation (e.g. 2c-and beyond, sup. fig2)?

Answer: The reason that we included other mutants in Fig. 3g is to indicate the importance of S42 phosphorylation for BMAL1 transcriptional function. We actually used other mutants to explore the key residues for AKT-phosphorylation and found that the mutation of other serine residues has little effect on AKT phosphorylation of BMAL1 proteins. We showed one of these results in Fig. 2d in the revised manuscript.

e) Regarding the assertion in page 8: "these results suggest that phosphorylation of Ser 42 by insulin triggers BMAL1 to dissociate from DNA..."; it is recommended to write instead: "may trigger BMAL1 to dissociate", because the experiments presented are not sufficient to demonstrate that the less or the more BMAL1 on the chromatin is due to the variation in BMAL1 levels within the nucleus or to a change in its affinity for target sequences.

Answer: We thank for Reviewer2's insightful suggestion and have corrected this issue in our present manuscript (page 12, line 11).

f) Page 8: "However, the role of insulin signal in food entrainment resetting hepatic clock is still unknown." Minimally here the studies of restricted feeding and circadian gene expression done in streptozotocin-treated or other animal models of diabetes should be cited (Oishi et al., 2004; Hoffman et al., 2013; Tseng et al., 2015).

Answer: We have cited these papers in our revised manuscript (page 4, line 11; page 12, line 21).

g) Page 9: The authors mention a reduction in leptin rhythm amplitude. Why did

they measure leptin? Instead, or in addition to, they should have measured glucagon, which is known to affect *Bmal1* (Sun et al., JBC 2014). And discuss it.

Answer: The reason that we measured leptin levels is not because we thought leptin may affect *Bmal1* expression. In the food entrainment experiments, we noticed that RF significantly increases food intakes, the mouse stomach sizes and weights at ZT4, 8, and 12 in each day (Fig. S3c and S3d in the last version and Fig 5e and Supplementary Fig. 4e in the current one). As leptin is one of the major hormones to inhibit animal appetite, we measured its plasma levels to show that RF not only reverses the phases but also decreases the amplitudes of leptin levels to result in the increase of food intakes. In addition, we measured the glucagon levels around Day3-ZT16, ZT4 when insulin levels were strikingly changed by RF, and found no significant difference between AF and RF groups (Supplementary Fig. 4f), which may be due to short fasting periods and lots of food consumption before fasting.

h) Page 9: "Taken together, our results revealed an essential role of insulin in the regulation of BMAL1 transcriptional activity by determining its nuclear localization via AKT-mediated S42 phosphorylation in the liver under physiological conditions (sup. fig 4f), which plays an initiating and sustaining function in hepatic clock reset by food entrainment". This notion is not fully supported by studies in diabetic rats (Oishi et al., 2004). The authors should discuss the discrepancies. Additionally, it is known that BMAL also affects the capacity of AKT for being activated by insulin (Zhang et al., JBC 2014). It would be also interesting to discuss it.

Answer: We agree with Reviewer2's comments. With the new results, we think that insulin plays an important role in the initiation of hepatic clock reset by food entrainment but not sustaining the whole process. We have changed the description (page 2, line 12; page 15, line 18) and discussed this issue as Reviewer2 suggested (page 16, line 9). Briefly, we think that the depletion of insulin may only delay but not block the reset of hepatic clock, as other hormones (e.g. glucocorticoids) also affect the expression of *Bmal1* and other clock genes. Under circumstances without activating insulin signaling in the liver, for example STZ injection, the oscillation of nuclear BMAL1 accumulation should most likely follow the fluctuation of its total expression. Thus, RF should be still capable of resetting the former by reversing the latter through glucocorticoid signaling in the absence of insulin signaling activity, as proposed in the study by Oishi et al. We also discussed the issue of BMAL1-AKT (Zhang et al., JBC 2014) as Reviewer2 suggested (page 17, line 1).

i) Animals and Experimental Design. The *Akt2*^{-/-} mouse line was from Z. Yang (to give the complete name and institution). The Alb-Cre mouse line was from Y Liu; and the *Bmal1*^{fl/fl} mouse line was purchased from Jackson Laboratories: at which experiments were these animals included? Specify.

Answer: The *Akt2*^{-/-} mouse line was from Dr. Zhongzhou Yang [Laboratory of Heart and Disease Model, MOE (Ministry of Education) Key Laboratory for Model Animal and Disease Study, Model Animal Research Institute, Nanjing University, Nanjing, China]; The Alb-Cre mouse line was generously provided by Dr. Yong Liu (Institute

for Nutritional Sciences, China); The *Bmal1^{fl/fl}* mouse line was purchased from Jackson Laboratories. The *Bmal1^{fl/fl}* mouse line was crossed with Alb-Cre line to generate liver specific Bmal1 knockout mice, which were used in the experiments of Fig 1g, and Supplementary Fig. 3i. We included the above information in the Method (page 18, line 3).

j) Cell culture. Given the importance of culture conditions for circadian clock gene expression (Yamajuku et al. Sci Rep 2012), the authors have to be more explicit in their methods.

Answer: HEK293T and HepG2 cells were purchased from ATCC, cultured in DMEM supplemented with 10% fetal bovine serum, 1% penicillin/streptomycin and 1% l-glutamine. Mouse primary hepatocytes were prepared, cultured and infected with adenoviruses as previously described⁵. Briefly, the hepatocytes were isolated using collagenase (Sigma) and plated on dishes coated with rat tail tendon collagen (Sigma) according to manufacturer's instructions. Before experiments, all the cells were synchronized by 12 hour serum starvation, which also reduced the background of insulin signaling. We have included the above information in the Method (page 19, line 19).

k) Page 12, Immunoblot: in which experiment was the HSP90 antibody used? It is needed the catalogue number of all the antibodies used, for reproducibility purposes. Page 13, Mass spectrometry: absolutely scarce, how did they prepared the samples, which mass spectrometry method and/or equipment was utilized? Which criteria did they use to characterize the phosphorylation sites?

Immunostaining. Important, specify after how many hours after serum synchronization were the experiments done.

Answer: There's no experiment using HSP90 antibody in our study. We're sorry for this mistake and have corrected it. We added the detailed information of all the antibodies used and mass spectrometry protocol in the Method (page 21, line 2; page 23, line 1). We synchronized hepatocytes with 12 hour serum starvation for immunostaining experiments.

l) The experiments done in vivo, with AKT2 KO animals (Figures 2g and 4b) would be suitable complemented (and the conclusions strengthened) using three animals injected in the tail-vein with Ad-FLAG-S42A-BMAL1 (the authors already have the adenoviral construction, as indicated in Fig 3H legend).

Answer: We have performed the experiment as Reviewer2 suggested and added the results in the revised manuscript (Supplementary Fig. 3d).

m) Fig. 3h. There are required control no-refed groups, in order to compare the changes in BMAL phosphorylation in response to a physiological challenge. Otherwise, how could be explained the abrupt change in DBP and NR1D1 proteins at just 30 minutes after re-feeding?

Answer: According to Reviewer2's suggestion, we have redesigned this experiment

by replacing refeeding with insulin injection and replaced the results in the revised manuscript (Fig. 4d).

n) Fig 1d. The subtitle "fasted" should be instead "time post-injection"

Answer: We're sorry for this mistake and have corrected it (Fig. 1d).

o) Fig 2c. Why the authors did not use an antibody to p-AKT-Substrate as in 2a and 2b, in order to know if the other two putative AKT phosphorylation sites in BMAL1 protein are also phosphorylated by insulin stimulation?

Answer: According to Reviewer2's suggestion, we have redesigned this experiment and added the result in the revised manuscript (Fig. 2d).

p) Fig. 2f. In this, as in Sup. fig. 2c it is suggested a Coomassie staining, to ensure the amount of protein is similar, and to ensure that it didn't have a proteolytic event in the protocol process. To include in Methods sf9 cells.

Answer: According to Reviewer2's suggestion, we have included the coomassie staining results in the revised version (Fig. 2g and Supplementary Fig. 2d).

q) Fig. 2g. Do AKT2 KO animals have a food intake circadian rhythm similar to WT? It is important to indicate that in the text.

In order to confirm the hypothesis that akt2 signaling is essential for nuclear BMAL1 circadian rhythm, it would be useful to compare the results with those of liver-specific constitutive-active AKT2 mutant mice.

Answer: We have measured the food intake circadian rhythm of *Akt2* KO and WT mice and found that *Akt2* KO mice have a food intake circadian rhythm similar to WT ones. The data are added in the revised version (Supplementary Fig. 3c).

As liver-specific constitutive-active AKT2 mutant mice are not available, we checked the effects of constitutive-active AKT2 mutant on nuclear BMAL1 circadian rhythm in primary hepatocytes. The results showed that constitutive-active AKT2 mutant (AKT2-CA) almost abolishes the oscillation of nuclear BMAL1 amounts (Fig. 3b and Supplementary Fig. 3e), which further supports our hypothesis.

r) Fig. 3f. Why are the input signals so weak?

Answer: The weak input signals are due to a short exposure time of the film during immunoblotting assay. We have replaced this weak input signals with a stronger one with a longer exposure time (Supplementary Fig. 3j).

s) Fig. 3g. I do not understand why Per1 transcription is not affected when the S422A and S513 variants are added conjointly with the S42A mutant (black bars)? Were they used in equal quantities? One could expect a decrease in the transcription rate due to competition; with a consequent bar length between the one of WT+CLOCK and S42A+CLOCK.

Answer: In the results of Fig. 3g, we did not use three plasmids with a single mutation in BMAL1-S42A/S422A/S513 experimental condition. Instead, we

transfected cells with only one plasmid expressing BMAL1 with a triple mutation and applied similar strategy for other conditions. Therefore, we used the same amount of BMAL1 plasmid in each condition. We changed the label and legend to clarify this issue (Fig. 4b in the revised manuscript).

t) Do AKT2 KO mice have no circadian rhythms in the clock-output genes?

Answer: We checked the expression profile of clock-output genes (e.g. *Dbp* and *Nr1d1*) in the liver of ad libitum fed AKT2 KO mice. The results showed that *Akt2* knockout enhances the expression levels of these two genes at multiple time points, especially during nighttime (Supplementary Fig. 3b in the revised manuscript), which is consistent with the increase of nuclear BMAL1 accumulation in these mice (Fig 3a in the revised manuscript). However, the expression patterns of these two genes still show daily rhythm with the similar phase in *Akt2* knockout mice, compared with WT ones, which may be due to the presence of other unaffected circadian regulators, such as CRY or glucocorticoid signaling).

u) Fig. Suppl.1: Histone and beta-actin groups are interchanged regarding their nuclear or whole localization.

Answer: We're sorry for this mistake and have corrected it (Fig S1a in the revised manuscript).

v) Fig. Suppl.2a: Indicate the m/z where is located the phosphorylation related residue, and indicate it in the diagram. The authors should show a similar figure for peptides containing the other two possible phosphorylation sites of BMAL1.

Answer: As Reviewer2 suggested, we have indicated the phosphorylation site in the diagram [Fig. S2a (up) in the revised manuscript]. When the company [Shanghai Applied Protein Technology Co. Ltd (China)] provided us the mass spectrum data in 2011, they only gave a list of all detected peptides with or without phosphorylation, which includes the other two possible phosphorylation sites of BMAL1 (e.g. S412 and S533, please see the below list) and details of the peptide containing S42 (the positive result), but not those unphosphorylated ones that including these two possible phosphorylation sites. As they have not kept the details for negative results, we cannot draw similar figures for these two possible phosphorylation sites. However, our other data (Fig. 2d in the revised manuscript) confirmed that these two sites are not phosphorylated by AKT.

Serine Residue #	Reference	PepCount	UniquePepCount	CoverPercent	MW	PI	IdentifiedName				GroupCount	ProteinCount	
			MH+	Diff(MH+)	Charge	Rank	KC	DeltaCn	Sp	RSp			Ions
Ser513	Bmal_T_8010	R.GSSPSSCOSSLNITSTPRPDASSPQGGK	2630.7544	0.1714	2	1	2.8741	1	325.9	1	1754	5.83	1
Ser513	Bmal_T_8015	R.GSSPSSCOSSLNITSTPRPDASSPQGGK	2630.7544	2.0684	2	1	2.4701	0.9247	341.5	1	1854	5.83	1
Ser513	Bmal_T_8030	R.GSSPSSCOSSLNITSTPRPDASSPQGGK	2630.7544	1.9914	2	1	3.0282	1	289	1	1854	5.83	1
Ser422	Bmal_T_12914	R.SRVFDFMNPVTK.E	1587.8287	0.4287	2	1	2.4885	0.8481	668	1	1322	11	1
Ser422	Bmal_T_12930	R.SRVFDFMNPVTK.E	1587.8287	1.1197	2	1	2.3976	0.8843	1262	1	1822	11	1

w) Fig. Suppl. 2c: Why is the Flag signal so weak in the WT?

Answer: It was due to a short exposure time, we have selected a stronger signal to replace the former one (Supplementary Fig. 2d in the revised manuscript).

x) Fig Suppl. 3f: Why is insulin-degrading-enzyme (IDE) here?

Answer: We're sorry for this mistake and have deleted it in the present version (Fig. 6d).

Reviewer #3:

(1) Studies in Akt2 KO mice support the concept that Akt 2 plays an important role in promoting nuclear exclusion of BMAL1, but they do not demonstrate that S42 is critical for this effect. Additional studies are needed to show that S42 is required to mediate effects of Akt on BMAL1 nuclear/cytoplasmic trafficking, and regulation of clock function. One experiment would be to express BMAL1-GFP fusion proteins with/without mutation of S24 and show that replacement of S24 with alanine disrupts the ability of insulin to promote nuclear exclusion of BMAL-GFP in an Akt-dependent fashion (e.g. with/without Akt inhibition). Co-transfection studies with constitutively active Akt also would demonstrate direct S42-dependent effects on BMAL1 translocation.

Answer: We have performed both experiments as Reviewer3 suggested. The data showed that BMAL1 with S42A mutation resists to insulin or constitutively active AKT (AKT-CA) induced nuclear exclusion (Fig 3f. and 3g in the revised manuscript).

(2) Similarly, studies are needed to determine whether S42 phosphorylation is required to mediate effects of insulin on the expression of downstream targets of BMAL1 (e.g. *Dbp*). If this is the case, overexpression of S42A BMAL1 would be expected to disrupt the ability of insulin (and/or Akt) to suppress the expression of BMAL1 target genes. Similar studies can be performed utilizing promoter assays to show whether BMAL1 protein and cis-acting BMAL1 sites are required to mediate effects of insulin/Akt on promoter activity, and whether overexpression of S42A BMAL1 blocks these effects of insulin/Akt.

Answer: We have performed the experiments according to Reviewer3's suggestion. As expected, these results showed that BMAL1-S42A disrupts the ability of insulin to suppress *Dbp* and *Nr1d1* expression (Fig. 4c-d in the revised manuscript)

(3) In vivo studies addressing the role of S42 phosphorylation in the regulation of clock function in the liver also would strengthen the study. Short of creating an S42A knock-in mouse, adenoviral expression of S42A BMAL1 in the liver would be expected to result in constitutively nuclear BMAL1, and disrupt both the regulation of clock in the liver, and the ability to reset the hepatic clock altering the timing of feeding. If these experiments succeed, they will provide clear evidence for S42 phosphorylation in the regulation of hepatic clock activity. If they do not show this result, they will suggest that other mechanisms are sufficient to maintain effects of refeeding time on clock activity in the liver.

Answer: We have performed these experiments according to Reviewer3's suggestion. As expected, our results showed that adenoviral expression of FLAG-BMAL1-S42A in the liver increases nuclear FLAG-BMAL1 accumulation, enhances clock controlled gene expression (e.g. *Dbp* and *Nr1d1*), especially under refeeding or insulin-injection conditions. (Fig. 6b-c and Fig. 4d in the revised manuscript). Consistent with the results of *Akt2* KO mice (Fig. 3a, Supplementary Fig 3b in the revised manuscript), the expression levels of hepatic *Dbp* and *Nr1d1* are increased at multiple time points in mice adenoviral expressing FLAG-BMAL1-S42A during Day1 of RF, especially in

the daytime (feeding period), but their daily expression patterns still exhibit similar fluctuations as those in mice with adenoviral expressing FLAG-BMAL1-WT (Fig. 6c in the revised manuscript), which may be due to either the existence of endogenous BMAL1 protein that is sensitive to insulin signaling, or the presence of other unaffected circadian regulators, such as CRY or glucocorticoid signal.

(4) The authors utilize a site-specific antibody to measure phosphorylation of serine 42 in BMAL1. Since this appears to be a new reagent that has not been previously reported, please describe how this antibody was made, and provide data demonstrating its specificity, including full length western blots and competitive inhibition studies with phospho- and non-phosphopeptides.

Answer: As Reviewer3 suggested, we have performed the experiments and added the data in the revised version (Supplementary Fig. 2e in the revised manuscript).

(5) p. 3, last sentence. The authors state several times that insulin "stimulates" translocation of BMAL1 from the nucleus to the cytoplasmic compartments. Since S42 is located within a nuclear localization signal, a more likely scenario is that phosphorylation of S42 masks this NLS, thereby preventing translocation from the cytoplasmic compartment into the nucleus, thereby trapping it in the extranuclear space. The text and summary figure should be revised to include this possibility.

Answer: We totally agree with Reviewer3's opinion and modified the corresponding description in the text (page 6, line2 in the current manuscript, corresponding to page 3, last sentence in the last submitted version) and summary figure (page 37, last sentence and Fig. 6d in the revised manuscript).

(6) p. 4, line 6. Remove the "s" at the end of "accumulations".

Answer: We're sorry for this mistake and have corrected it in the revised manuscript (page 6, line 9).

(7) p. 4, last line. Suggest replacing "determine" with "limit".

Answer: We corrected this issue in the revised manuscript (page 7, line 5).

(8) p. 5, line 4. Insert "of" after "exposure".

Answer: We're sorry for this mistake and have corrected it in the revised manuscript (page 7, line 12).

(9) p. 6 and Fig 2g.

Answer: We're sorry not to understand the question, but assume that Reviewer3 may think that the description of "resulting in the loss of its nuclear rhythm (fig. 2g)" (page 6, line 2 in the last version) is unappreciated. Therefore, we changed it to "which results in the disruption of its nuclear rhythm (Fig. 3a)", (page 9, line 10) in the revised manuscript

(10) The authors note that S42A BMAL1 is more efficiently recruited to the Dbp and

Nrfd1 promoters, and activate the Per1 promoter, and suggest that this reflects improved DNA binding. However, It seems more likely that this may simply reflect differences in the level of BMAL1 protein (S42A >> WT) in the nucleus, and not differences in DNA binding efficiency. To show that there are differences in DNA binding efficiency, additional studies would be needed to assess binding activity under conditions where WT and S42A protein levels are comparable - e.g. in gel shift assays where levels of WT and S42 BMAL1 can be controlled. (Note: if differences are observed, 14-3-3 proteins may contribute to this effect.). Without this data, the authors need to adjust the text to allow for the possibility/likelihood that nuclear exclusion of WT (but not S42A) BMAL1 contributes to differences in transcriptional activity.

Answer: We totally agree with Reviewer3's concern. Although we didn't perform the gel shift experiments as suggested by Reviewer3, we checked the nuclear amounts of BMAL1-WT and -S42A proteins in the experiments of Fig. 3e and Fig. 3g in the last submitted manuscript, and found that that nuclear BMAL1-S42A protein levels are mildly less than its WT controls under the basal condition without insulin treatment, if infected hepatocytes [Fig. 4a (right), in the revised manuscript] or transfected HEK293T [Fig. 4b (bottom), in the revised manuscript] with the same amounts of adenoviruses or plasmids of BMAL1-WT or S42A. Together, our results showed that BMAL1-S42A has higher occupancy on target gene promoters and induction of Per1-Luc activity than BMAL1-WT, even with less nuclear protein amounts, which suggests that BMAL1-S42A protein has higher affinity to DNA than WT.

(11)fig 3 f. The FLAG western blot for protein input is weak. Can the authors provide a more convincing western?

Answer: We have replaced that weak input with a stronger one in the present version (Supplementary Fig. 3j in the revised manuscript).

(12) sup fig 3. What is "AF rep"?

Answer: Leptin levels of ad libitum feeding group presented in Day2 and 3 were not directly measured, instead replicated from those of Day1 (assumedly no significant change among different days under ad libitum condition). Thus, we represented it as "AF rep". We have annotated it in the figure legend of the revised manuscript.

(13) sup fig 3f. What is the significance of IDE? This is not mentioned in the text. Please explain, or delete.

Answer: We're sorry for this mistake and have deleted it in the present version (Fig. 6d in the revised manuscript).

Reference:

1. Yuji Shi. *et al.* PTEN is a protein tyrosine phosphatase for IRS1. *Nat Struct Mol Biol.* **21**, 522-527 (2014).
2. Pengda Liu. *et al.* Sin1 phosphorylation impairs mTORC2 complex integrity and inhibits downstream Akt signalling to suppress tumorigenesis. *Nat Cell Biol.* **15**,

1340-1350 (2013).

3. Rey, G. *et al.* Genome-wide and phase-specific DNA-binding rhythms of BMAL1 control circadian output functions in mouse liver. *PLoS Bio.* **9**, e1000595 (2011).
4. Lamia, K. A., Storch, K. F. & Weitz, C. J. Physiological significance of a peripheral tissue circadian clock. *Proceedings of the National Academy of Sciences of the United States of America.* **105**, 15172-15177 (2008).
5. Dentin, R. *et al.* Insulin modulates gluconeogenesis by inhibition of thecoactivator TORC2. *Nature.* **449**, 366-369 (2007).

REVIEWERS' COMMENTS:

Reviewer #1 (Remarks to the Author):

The authors have addressed my concerns although still many blots have not been properly quantified. The language could still be improved.

Reviewer #2 (Remarks to the Author):

I would like to complement the authors for the extensive revision and additional experiments they have done. Now the provided evidence for insulin mediated phosphorylation of BMAL1 is convincing and interesting and may provide an explanation for the changed rhythm of BMAL1 under reversed feeding conditions.

Reviewer #3 (Remarks to the Author):

The authors have addressed the major points raised by this reviewer through the performance of additional studies.

The following additional point emerge that require revision:

1. The authors find that overexpression of an alanine mutant partially alters, but does not completely disrupt the diurnal regulation of several transcription, but not others. This result confirms that phosphorylation at this site contributes to, but does not fully account for the diurnal regulation of gene expression in the liver in response to changes in insulin levels. that are under diurnal control, and that other mechanisms also are involved. This needs to be clearly stated in the results section, and discussed, since it indicates that multiple mechanisms contribute to the diurnal regulation of gene expression in the liver, as might be expected - and not just Akt phosphorylation of BMAL1.
2. Related - the title of the manuscript should be modified. As it stands, it suggests that phosphorylation of BMAL1 is the major mechanism mediating effects of insulin on the hepatic circadian clock. This is not established. Instead, the authors have done an excellent job identifying phosphorylation by Akt in regulating nuclear-cytoplasmic trafficking of BMAL1, and showing that this contributes to (but does not fully account) for the regulation of target genes.

Reviewer #3 (Remarks to the Author):

1. The authors find that overexpression of an alanine mutant partially alters, but does not completely disrupt the diurnal regulation of several transcription, but not others. This result confirms that phosphorylation at this site contributes to, but does not fully account for the diurnal regulation of gene expression in the liver in response to changes in insulin levels. that are under diurnal control, and that other mechanisms also are involved. This needs to be clearly stated in the results section, and discussed, since it indicates that multiple mechanisms contribute to the diurnal regulation of gene expression in the liver, as might be expected - and not just Akt phosphorylation of BMAL1.

Answer: We completely agreed with Reviewer3's comments and added the corresponding description in the results (page 14, line 18) and discussion (page 16, line 1).

2. Related - the title of the manuscript should be modified. As it stands, it suggests that phosphorylation of BMAL1 is the major mechanism mediating effects of insulin on the hepatic circadian clock. This is not established. Instead, the authors have done an excellent job identifying phosphorylation by Akt in regulating nuclear-cytoplasmic trafficking of BMAL1, and showing that this contributes to (but does not fully account) for the regulation of target genes.

Answer: We changed the title according to Reviewer3's suggestion.